



# Evaluation of Aeris MIRA, Picarro CRDS G2307, and DNPH-based sampling for long-term formaldehyde monitoring efforts

Asher P. Mouat[1], Zelda A. Siegel[1], Jennifer Kaiser[1,2]

[1]School of Civil and Environmental Engineering, Georgia Institute of Technology, Atlanta, Georgia 30332, USA
[2]School of Earth and Atmospheric Sciences, Georgia Institute of Technology, Atlanta, Georgia 30332, USA

*Correspondence to*: Jennifer Kaiser (Jennifer.kaiser@ce.gatech.edu)

**Abstract.**

Current formaldehyde measurement networks rely on the TO-11A offline chemical derivatization technique, which can be resource intensive and limited in temporal resolution. In this work, we evaluate the field performance of three new commercial instruments for continuous in-situ formaldehyde monitoring: the Picarro cavity ringdown spectroscopy (CRDS) G2307 gas concentration analyzer and Aeris Technologies' mid-infrared absorption (MIRA) Pico and Ultra gas analyzers. All instruments require regular drift correction, with baseline drifts over a 1-week period
of ambient sampling of 1 ppb, 4 ppb, and 20 ppb for the G2307, Ultra, and Pico, respectively. Baseline drifts are easily corrected with frequent instrument zeroing using DNPH scrubbers, while Drierite, molecular sieves, and heated hopcalite fail to remove all incoming HCHO. Drift-corrected $3\sigma$ limits of detection (LOD) determined from regular instrument zeroing were relatively comparable at 0.055 ppb (Picarro G2307), 0.065 ppb (Aeris Ultra), and 0.08 ppb (Aeris Pico) for a 20 min integration time. We find that after correcting for a 30-40% bias in the Pico measurements,
all instruments agree within 5% and are well correlated with each other (all $R^2 \geq 0.70$). Picarro G2307 HCHO observations are more than 50% higher than co-located TO-11A HCHO measurements ($R^2 = 0.92$, slope = 1.47, int = 1 ppb HCHO), which is in contrast to previous comparisons where measurements were biased low by 1–2 ppb. We attribute this discrepancy to the previous versions of the spectral fitting algorithm as well as the zeroing method. The temperature stabilization upgrade of the Ultra offers improved baseline stability over the previously described Pico
version, reducing the maximum drift rate by a factor of 13 and improves precision of a 10 min average by 13 ppt. Using a 6-month deployment period, we demonstrate that all instruments provide a reliable measurement of ambient HCHO concentrations in an urban environment and, when compared with previous observations, find that midday summertime HCHO concentrations have reduced by approximately 50% in the last two decades.

## 1 Introduction


Observations of formaldehyde (HCHO) provide useful insight into the photochemical formation of secondary pollutants and the sources and fate of volatile organic compounds (VOCs). While direct emissions of HCHO from wildfires, the biosphere, and anthropogenic activities can contribute to ambient mixing ratios (Parrish et al., 2012; Lui





et al., 2017; Luecken et al., 2018; Alvarado et al., 2020; Wu et al., 2021), regional HCHO abundance is generally
governed by secondary production (Parrish et al., 2012; Zhang et al., 2013; Zhu et al., 2014; Luecken et al., 2018; Zeng et al., 2019). Because HCHO is a source of $HO_X$ radicals, HCHO loss can further propagate oxidative chemistry (Tonnesen and Dennis, 2000; Lin et al., 2012; Valin et al., 2016; Wolfe et al., 2019; Yang et al., 2021) Additionally, HCHO is a known carcinogen ranking highest in health risks among the 187 hazardous air pollutants listed by the US Environmental Protection Agency (EPA) in the Clean Air Act (Scheffe et al., 2016; Strum and Scheffe, 2016; Zhu et
al., 2017b). Due to its central role in atmospheric chemistry, HCHO observations are a target molecule at EPA Photochemical Assessment Monitoring Station (PAMS) and National Air Toxics Trends Station (NATTS) network sites, and are typically included in chemically comprehensive field intensives.

Since 1990, the EPA-standard approach for HCHO measurements is collection on 2,4-dinitrophenylhydrazine
(DNPH) coated cartridges followed by offline derivative detection via high performance liquid chromatography (HPLC), known as the TO-11A method (Riggin, 1984). Sample collection and analysis is resource and labor intensive with measurements typically reported over long sampling times. EPA TO-11A measurements are 8 or 24 h integrated samples collected every three or six days, respectively. The low time resolution limits the usefulness of observations for studies of both photochemistry and air toxics exposure. Previous approaches have used modelled diel cycles or
satellite-based observations in combination with the TO-11A method to infer ground-based diel cycles (Zhu et al., 2017a; Zhu et al., 2017b; Wang et al., 2022). However, the method has known interferences from $NO_2$ and $O_3$ (Karst et al., 1993; Achatz et al., 1999; Tang et al., 2004), is not suitable for sampling in conditions with low relative humidities (Wisthaler et al., 2008; Uchiyama et al., 2009), and has had mixed results in comparison to research-grade observations (Hak et al., 2005; Dunne et al., 2018), making the accuracy of these inferred diel cycles difficult to
determine. While other studies have demonstrated the feasibility for continuous measurements via various spectroscopy-based methods (Yokelson et al., 1999; Cardenas et al., 2000; Dasgupta et al., 2005; Hak et al., 2005; Spinei et al., 2018; St Clair et al., 2019; Dugheri et al., 2021), the number of long-term (longer than 1 month), ground-based, continuous HCHO measurements is limited to a handful of studies, all of which employ either the multi-axial differential optical absorption spectroscopy measurement technique (Tian et al., 2018; Kumar et al., 2020; Hoque et
al., 2022) or a proton-transfer-reaction mass spectrometer, for which HCHO measurements are sensitive to humidity fluctuations (Warneke et al., 2013; Hansen et al., 2014).

A more suitable long-term HCHO monitoring instrument would reduce manual labor and required knowledge for operation, provide continuous observations, experience little or correctable drift in instrument baseline and sensitivity,
and have low uncertainty and sufficient precision at typical ambient concentrations. In recent years, several commercially available instruments have been developed towards that goal, including a cavity ring down spectroscopy (CRDS) instrument from Picarro, a photoacoustic gas analyser from Gasera, and Tunable Diode Laser Spectroscopy (TDLS) instruments from Aeris Technologies and Aerodyne Research, Inc. Here, we focus on the Aeris MIRA and Picarro CRDS G2307 instruments, which have been compared against other instruments in a small number of informal
(Whitehill et al., 2018; Furdyna, 2020) and peer-reviewed (Shutter et al., 2019; Glowania et al., 2021) intercomparison



efforts. Since those efforts, the Picarro G2307 instrument implemented updates to its spectral fitting algorithm, and the Aeris MIRA instrument has offered improved thermal stabilization. Whereas previous comparisons were conducted either in controlled chamber studies or through analysis of short-term ambient observations, a full characterization of instrument suitability in measurement networks requires long-term deployment.


Previous intercomparisons involving either Aeris MIRA or Picarro CRDS instruments have highlighted concerns with measurement accuracy as a function of ambient humidity. The Aeris MIRA technique relies on the HDO line for spectral referencing. At low humidity (< 2000 ppm $H_2O$), the Aeris Real-Time (ART) fitting algorithm cannot reliably reference the HDO spectral feature and the instrument fails to produce measurements (Shutter et al., 2019). Including

$CH_4$ as a secondary spectral reference in data post-processing extends the range of conditions under which the Aeris instruments work, though the instrument's precision decreases by a factor of $1.2 \pm 0.3$. While the G2307 can make use of both $H_2O$ and $CH_4$ references, this currently remains a research approach for ART. Whitehill et al. (2018) found an inverse correlation between Picarro HCHO measurements and instrument-reported water mixing ratios at typical, ambient concentrations and, along with Furdyna (2020), observed that the G2307's measurements were lower by 1-2

ppb HCHO compared to DNPH-based measurements. Glowania et al. (2021), using a spectral fitting algorithm updated after the Whitehill et al. (2018) intercomparison (released Sep. 2019), found that low humidity conditions can lead to changes in reported HCHO concentrations as high as 1.75 ppb. These offsets are most significant at $\leq 0.2\%$ $H_2O$ where the $H_2O$ spectral feature is not clearly observed. This updated algorithm was designed as an improvement to the spectral fitting procedure in low-humidity conditions.


Both instruments rely on periodic instrument baseline zeroing by sampling HCHO-free air. Several scrubbers are capable of removing HCHO – the most common of which are DNPH-coated cartridges, heated oxides of copper and manganese (hopcalite, HO), calcium sulfate (Drierite, DR), and molecular sieves (MS) (Cazorla et al., 2015; Pei et al., 2015; Shutter et al., 2019; St Clair et al., 2019; Fried et al., 2020). These methods differ in removal mechanism,

molecular selectivity, and desiccation efficiency. DNPH-coated cartridges are recommended by Aeris Technologies, and are chemically selective for carbonyls, thus allowing the majority of $H_2O$ to pass through. Heated HO is expected to oxidize HCHO to CO, forming $H_2O$ as a by-product and providing a humidified airstream that may also be suitable for baseline determination. Picarro Inc. recommends instrument zeroing via adsorption by DR. A column of MS is often plumbed in upstream of a DR column (DR+MS) as it both desiccates the gas flowing through it and, with the

right pore size, removes molecules with kinetic diameters greater than that of HCHO. This both prevents the DR from becoming saturated and prolongs its HCHO-removal efficiency as only smaller organic compounds can interact with it. HO and DR+MS may be less cost-intensive, longer-lasting, and have comparable HCHO-removal efficiency to DNPH-coated cartridges. As humidity is previously known to impact HCHO concentrations, impact of scrubber on overall measurement accuracy is unclear.


We use long-term HCHO measurements in Atlanta, GA from the Picarro G2307 and the Aeris instruments with aims to determine an optimal measurement configuration and assess suitability for remote continuous deployment. We



compare co-located observations from (1) Picarro G2307 and Aeris Pico, (2) the two Aeris instruments (Ultra and Pico), and (3) the Picarro G2307 and TO-11A measurements. For each instrument, we assess the performance over a range of zeroing methods and ambient humidities. Finally, we demonstrate the use of Picarro G2307 and Aeris Ultra and Pico measurements for long-term, continuous observations of HCHO spatial gradients in an urban environment and discuss the feasibility of deploying these instruments to form a spatiotemporally comprehensive network.

## 2. Instrument descriptions

### 2.1 Picarro G2307 description and calibration

The operating principle of cavity ringdown spectroscopy as used by the G2307 is described fully in Glowania et al. (2021), and briefly summarized here. Air is pulled through a temperature and pressure-controlled cavity at a rate of 0.4 standard liters per minute (SLPM). Laser light is directed into the resonance cavity, where three high-reflectivity mirrors create effective pathlengths on the kilometer scale. After the laser is shut off, the small amount of light transmitted through one mirror is monitored via photodetector. Detected light exponentially decays, with faster decay rates corresponding to higher absorption of light in the cavity. An on-board wavelength monitor measures the absolute laser wavelength with a precision that is three order of magnitude narrower than the HCHO spectral linewidth. The instrument can change the voltage applied to the laser and tune it to wavelengths that HCHO is known to either minimally or maximally absorb at, producing closely clustered spectral features at and around the HCHO absorption peak. The laser scans the 5625.5 to 5626.5 cm$^{-1}$ wavelength range at 100 Hz repetition rate, while the length of the cavity is adjusted to achieve resonance. On-board spectral fitting and signal averaging results in measurements of HCHO, $CH_4$, and $H_2O$ reported at 1 Hz.

The G2307 measurements reported here differ primarily in that we employ an external zeroing system equipped to sample from DNPH-coated cartridges (Sigma Aldrich LpDNPH S10L), DR (Drierite, 8 mesh, >98% CaSO4, <2% CoCl2), or DR+MS (Sigma Aldrich Molecular Sieve, 0.3 nm zeolite beads) to regularly monitor and correct for instrument drift (shown in Fig. 2a). This was accomplished by connecting the G2307 inlet to a 3-way PFA solenoid valve which alternated between the ambient sampling line and a zeroing line. The zeroing line was then connected to another 3-way PFA solenoid valve to which the scrubbers were attached. The instruments sampled from DR or DR+MS for 5 min of every hour. Every fourth hour, the instrument sampled for 5 min from the DNPH line either directly before or after sampling from DR. The relative order of DR/DNPH sampling was found to have no impact on reported instrument baselines.

We create averaged HCHO datasets at variable time resolutions (1 – 60 min) from the 1 Hz data using the following data processing procedure: All data taken within 30 s of a valve change are removed and the remaining 4.5 min of zero data is averaged to a single point. The zeros are linearly interpolated to create an instrument background on the same time basis as ambient data. The interpolated baseline is subtracted from the 1 Hz ambient measurements.



Baseline-corrected ambient data is averaged to the desired time resolution with any periods missing ≥50 % data completeness discarded. Data is further screened to exclude points where instrument baselines are unreliable due to breakthrough or saturation in the scrubbers.

Instrument calibrations were performed before the measurement period using a cylinder containing 1.019 ppm ± 5% of HCHO in $N_2$ (Apel-Riemer) and at the end of the measurement period using a cylinder with 1.031 ppm ± 10% of HCHO in $N_2$ (Airgas). We determined instrument sensitivity using two points: a zero, determined from scrubbed air, and a point sampling directly from the tank. Instrument response is assumed to be linear in this range. This setup avoids interactions between the calibration gas and mass flow controllers (MFCs). While the standard dilution-method

calibrations using a HCHO cylinder and synthetic air showed linearity at ambient levels (1-10 ppb HCHO), measured concentrations were consistently 7% lower than expected, likely due to long timescales of surface passivation for HCHO in the MFC. The calibrations from the two cylinders were in good agreement (sensitivities within 2.5 %), suggesting no change in instrument sensitivity during our measurement period. We apply the sensitivity derived from the Apel-Riemer cylinder to all data shown here. Measurement uncertainties in the analyses below are assumed to be

equivalent to the manufacturer reported values (10%).

### 2.2 Aeris Pico and Ultra MIRA Instrument description and standard addition tests

The operating principle of the Aeris MIRA instruments is described fully in Shutter et al. (2019). Air is pulled air at a rate of 0.45 – 0.75 SLPM into a folded Herriott detection cell, which achieves a path length of 1.3 m. The laser scans over the HCHO feature at 2831.6413 $cm^{-1}$, as well as the nearby HDO spectral feature at 2831.8413 $cm^{-1}$. The ART algorithm corrects for broad slope in the raw signal of the instrument baseline, and then calculates measured HCHO and $H_2O$ concentrations based on absorption features. We use the two commercial Aeris MIRA models in this work:

the Pico and the newer Ultra model. The Ultra is identical in operation but offers higher optical cell temperature stability and is designed for longer-term, low-drift measurements.

The Aeris instruments have a two-inlet design allowing for determination of instrument baseline throughout the data collection process. We run the instruments in the "programmed" mode, which allows the user to select the duration of

sampling through each inlet. The instruments also have a "differential" mode, which produces ambient HCHO concentrations using on-board baseline subtractions. The zero line is connected to either a DNPH-coated cartridge or a heated HO (United Filtration) scrubber, and then to the main sampling line (Fig. 2b). Scrubbing ambient air rather than indoor air (as was done for the Picarro, Fig. 2a) ensures that $H_2O$ is present in scrubbed air, which is necessary for spectral referencing. We sample ambient air for 180 s and scrubbed air for 30 s. We generate temporally averaged

datasets with variable time resolutions (1–60 min) using a data handling scheme like that used to process Picarro G2307 observations (zeroes are averaged to single point, subtracted from the previous 1 Hz ambient periods, and ≥50 % data completeness is required). We discard the first 5 s of measurements after a valve switch.





Instrument calibrations with gas standards in $N_2$ prove difficult for the Aeris instruments as they require an addition
of $H_2O$ in the calibration mixture. Aeris instruments arrive pre-calibrated from the manufacturers with sensitivity
expected to be relatively stable. Factory calibration occurred roughly 15 months before the observations outlined here,
during which time the instruments had operated periodically under a range of conditions. We performed two
assessments of the factory calibration: the first was a standard addition test, in which HCHO from the Airgas standard
was added in sequentially greater concentrations to the sample line at flow rates less than the intake of the instrument.
The second was a cross-comparison of the Pico instrument with the calibrated Picarro G2307 instrument, discussed
fully in Sect. 4.

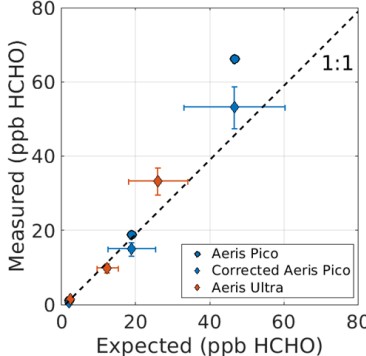

**Figure 1 – Results from Aeris standard additions. Each point is a 45 min average. Corrected Pico concentrations use the**
**Pico v. G2307 regression in Sect. 4. Error bars represent uncertainties in the measured and expected concentrations.**

Results from the standard addition calibrations are shown in Fig. 1. The Pico and the Ultra instruments were co-located
and sampling ambient air during the standard addition tests. Expected concentration was determined as the flow
weighted average of the gas standard and ambient concentrations measured by the other instrument. However, as
shown both in Fig. 1 and later in Sect. 4., the Pico consistently produced measurements with a high bias and indicated
that significant deviation in sensitivity had occurred since its last factory calibration. To reliably provide an ambient
reference point for the Ultra during its standard addition, all Pico observations were first corrected according to the
Pico v. G2307 regression shown in Fig. 7 of Sect. 4.

The uncertainty in all Pico and Ultra measured concentrations incorporate the (10% + 0.3) ppb instrument accuracy
determined in Shutter et al. (2019). The expected concentrations in the standard addition tests additionally incorporate
the 10% uncertainty in the standard concentration. Uncertainties introduced from the Pico v. G2307 regression,
instrument flow rate variability, and cylinder flow rate variability were all found to be negligible and were not
included. The Ultra's standard addition test shows that measured concentrations have a normalized mean bias (NMB)
of 9.01%, which is less than the uncertainty of the standard and our technique, so we conclude its sensitivity has



remained relatively stable over 15 months. Correcting the Pico's measured concentrations reduced the normalized mean bias from 27.8% to -1.98%.

**2.3 DNPH (TO-11A)**

Method TO-11A outlines in detail the EPA guidance on preparation of DNPH-coated cartridges and subsequent analysis through HPLC (Riggin, 1984). Formaldehyde was measured using an ATEC Model 8000 Toxic Air Sampler over three consecutive eight-hour periods spanning a full 24 hours with samples collected every three days. Ambient

air was drawn at a rate of $0.9 - 1.1$ L/min through a heated inlet and an $O_3$ denuder before passing through a DNPH-coated cartridge (Supelco DNPH-C-18) which collected carbonyls in their non-volatile, carbonyl-hydrazone derivative form. The denuder is necessary as it minimizes potential $O_3$-related interferences in the resultant HPLC chromatograms (Vairavamurthy et al., 1992). At the end of the sampling period, the cartridges were capped and stored in a refrigeration unit at $\leq 4$ °C until analysis. The cartridges were then eluted with 10 mL of acetonitrile (ACN) and

the eluent analysed via a Waters HPLC-UV system with a temperature stabilized ($25 \pm 1$°C), reversed phase C18-coated silica gel (1.7 µm particle size) column (Bridged ethyl hybrid, 2.1 mm x 50 mm ID) at 360 nm wavelength. The eluents used in the HPLC process were deionized $H_2O$ and ACN. The HPLC system was calibrated before each use with known concentrations of HCHO and field samples are analysed in comparison to blank cartridges.

Data here were reported to have minimum detection limits (MDLs) of $98 - 172$ ng (minimum signal-to-noise ratio of 3). The relative difference in mass collected between the primary and duplicate samples ranged from $-5.4 - 0.3$ % (n = 7 pairs of observations). We estimate a measurement uncertainty of 12 % by incorporating the average difference between duplicates and the allowed flow rate variability ($\pm 10$% of the design flow). This does not account for any biases caused by interfering species such as $NO_2$ (Karst et al., 1993).


**3 Field deployments**

**3.1 South DeKalb: Aeris Pico and Picarro G2307**


The locations of the South Dekalb (SDK) PAMS site and Georgia Tech (GT) field sites are shown in Fig. 2. The sites are approximately 12 mi apart with GT situated in the center of the city's urban core and SDK located in a less industrialized area with greater tree coverage. The G2307 and Pico instruments were co-deployed at SDK from 28 July to 13 Sept 2022 according to the configuration shown in Fig. 3a. Instruments were housed in a climate-controlled

trailer with an indoor temperature maintained at 21 °C. All tubing was 0.25 in OD (0.125 in ID) PTFE with 7.5 m extending from inside the trailer and up a mast, where the inlet was situated 5 m above the ground. Both instruments had a flow rate of 450 cm$^3$ min$^{-1}$, leading to a residence time of approximately 4 s. A 1µm particulate filter (PF) in a



Savillex holder was used, and the inlet was shielded by a PTFE funnel covered by PTFE mesh. The indoor portion of the sampling line was heated to 46 °C to avoid condensation in the sampling line.


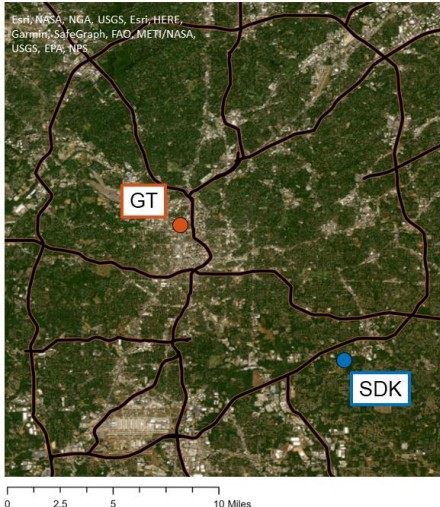

**Figure 2 – Locations of the two field sites in the Atlanta, GA area where the Aeris Ultra, Aeris Pico, and Picarro G2307 were deployed.**

The Aeris Pico baseline was determined using DNPH-coated cartridges while the Picarro G2307 sampled between DNPH, DR, or DR+MS. When scrubbing only with DR, air was passed through two adsorption columns (length of 16 in, radius of 2 in) in series containing 0.5 kg of material each. For DR+MS, the column first in series was replaced with the MS material. When the adsorption columns are exhausted, the scrubber bed is replaced with either new or regenerated material. DR is thermally regenerated according to the manufacturer instructions.


### 3.2 Georgia Tech: Aeris Pico and Ultra

The Aeris instruments were co-deployed in the penthouse laboratory of the Ford Environmental Science and
Technology building (GT) from 25 – 28 July 2022 and 4 – 18 Oct 2022 with the setups used during their co-located periods shown in Fig. 3b. Ambient temperature of the lab was maintained at 22 °C. A total of 7 m of 0.25 in OD (0.125 in ID) PTFE line ran from the instruments through a wall port, where the inlet was suspended 3 m above the outdoor roof floor. As before, a 1μm PF in a Savillex holder was attached and the inlet shielded with a PTFE funnel, and indoor tubing was insulated to prevent condensation from forming.


The Aeris Ultra solely used the DNPH scrubbing method for zeroing. The zeroing inlet on the Aeris Pico was teed to a DNPH-coated cartridge and a stainless-steel column (length of 8 in, radius of 0.75 in) containing 215 cm$^3$ of HO



(shown in Fig. 3b). The HO column was wrapped in high temperature heat tape, insulated in a fiberglass sleeve, and heated it to 180 °C. Pei et al. (2015) found HO at this temperature achieved nearly 100 % HCHO removal and

preserved the scrubber bed from $H_2O$ poisoning. A condensation trap and second PF are placed downstream of the HO column to protect the instrument against potential $H_2O$ formed in the HO catalyst during VOC oxidation. To compare the DNPH and HO scrubbing methods, two mass flow controllers were placed upstream of the scrubbers. The Aeris Pico sampled solely from its zeroing inlet while the sample flow alternated between scrubbers in 40 s intervals. A slight modification is made to the data processing scheme presented in Sect. 2.2 in that the first 10 s of

data after every switch is removed. This removal period was determined experimentally to fully exclude any sampling effects.

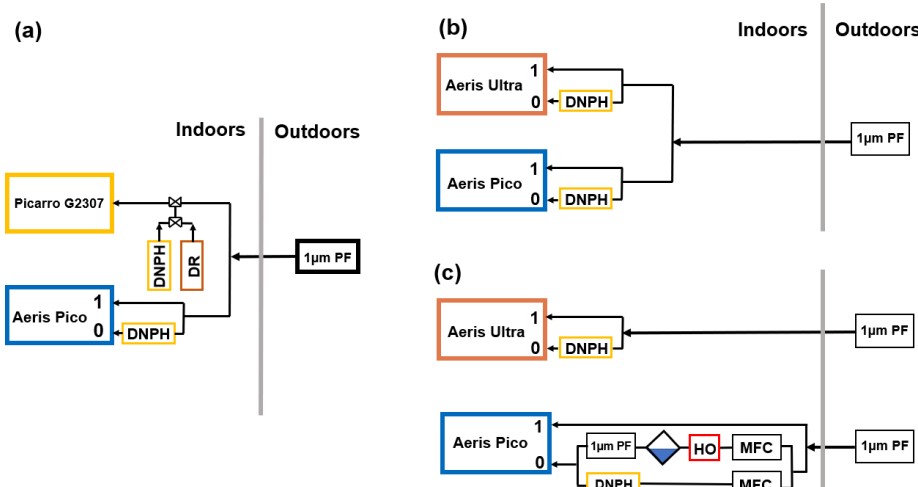

**Figure 3 – Configurations of instruments during their respective field deployments. (a) shows the teed setup used from 28**
**July – 13 Sep 2022 for the Aeris Pico and Picarro G2307. (b) and (c) shows the two setups used for the Aeris instruments while deployed at GT with the configuration in (b) used from 25 – 28 July and 4 Oct to present, and the configuration in (c) used to quantify differences in the HO and DNPH scrubbing methods.**

## 4. Results

### 4.1 Instrument precision and drift

Precision of the three analysers was characterized with Allen-Werle curves using instrument baselines measured while sampling through a DNPH-coated cartridge. This scrubber is chosen both because it is well regarded for use with absorption-based HCHO-sensing techniques and because it was the only zeroing method used across all three instruments assessed in this work. The G2307 valve sequencer was set to the same sampling schedule as the Aeris instruments (180 s sampling ambient, 30 s sampling scrubbed air) for one week, accumulating a non-continuous 24 h

period of scrubbed air. A field precision is calculated by correcting ambient data for instrument drift. The average of





each 25 s zeroing period is subtracted out from the 1 Hz baseline data, which are then treated as contiguous when calculating each instrument's Allen-Werle curve (Fig. 4).

The Picarro G2307 and the Aeris Ultra have comparable precisions for averaging windows between 5 – 20 min, with
the Aeris Pico exhibiting a slightly lower precision. The first local minima for the Aeris instruments occur at the 20 min averaging window with precisions at 0.065 ppb and 0.08 ppb for the Ultra and Pico, respectively. At the same averaging time, the Picarro G2307 achieves a precision of 0.055 ppb. Picarro Inc. report a precision of 0.06 ppb over 5 min for the G2307, for which we find our instrument performs closely reaching a value of 0.07 ppb over the same integration period. Shutter et al. (2019) determined an LOD of 0.42 ppb (equivalent to a precision of 0.14 ppb) over a
40 min integration period when using the ART fitting algorithm. The better precision in this work is owed to the drift-correction of the data, whereas Shutter et al. (2019) calculated a lab precision by flowing humid, ultra-zero air into their instrument for 20 h and using the uncorrected data. When integrating ambient data at different time resolutions (1-20 min), the Ultra produces precisions that are on average 95 ppt HCHO lower than the Pico.


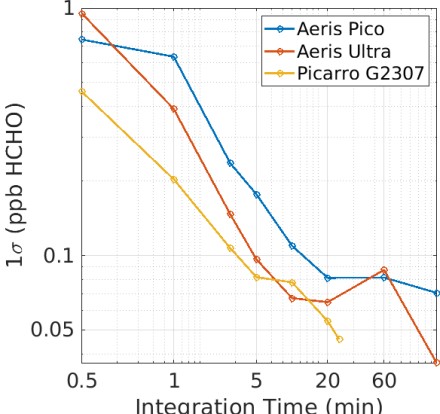

**Figure 4 – Allen-Werle curves for each of the instruments assessed in this work, derived using drift-corrected zeros from scrubbed, ambient air. The DNPH zeroing method is used for all instruments with baseline data taken over 1 week and treated as a contiguous 24 h period.**


To quantify instrument drift, we show a typical time series of scrubbed-air observations for all three instruments. The period chosen spans from 3 – 8 Sep 2022 (Aeris Pico and Picarro G2307 at SDK, Aeris Ultra at GT) and are shown in Fig. 5. The zero measurements are averaged according to the respective data scheme for each instrument and plotted differentially relative to the first value in each time series. The G2307 exhibits comparatively little drift with a max
difference of 1.1 ppb when sampling through either DNPH- or DR-scrubbed air, occurring late on 6 Sep. Over the same timeframe, the Aeris Ultra's baseline can shift up to ± 6 ppb while the Aeris Pico baseline exhibits the most variability, changing by as much as ± 20 ppb just over the course of 12 h. This significant drift is attributable to the lack of thermal stabilization in the instrument and necessitates more frequent zeroing, thus reducing total time spent



sampling ambiently and exhausting scrubbers faster. At their fastest drift rates (1.67 ppb HCHO h$^{-1}$ for the Pico and
0.125 ppb HCHO h$^{-1}$ for the Ultra), the improved thermal stability reduces drift by a factor of 13.36. From our
observations, we determined that the Pico should be zeroed at least every 6 min and the Ultra every 10 min under
typical indoor-deployment configurations. For the G2307, observations of the instrument baseline drift obtained using
DR suggest that hourly zeroing is sufficient.

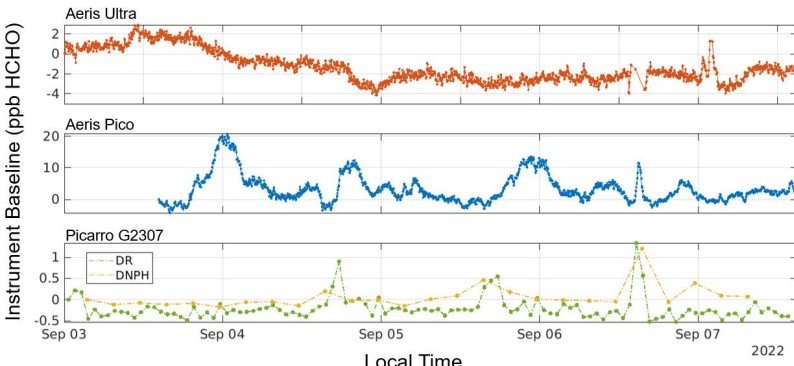


**Figure 5 – Instrument baseline time series for all three HCHO monitors plotted relative to the first point in the time series.**
**The Ultra and G2307, equipped with better thermal stabilization, show significantly less drift than the Pico.**

## 4.2 Impact of scrubber choice – DNPH, DR, and DR+MS for Picarro G2307


We assess the impact of DR and DR+MS on the Picarro G2307's baseline using measurements taken from every
fourth hour when the valve sequencer switched consecutively between DNPH and the alternative scrubber. We find
no significant difference when incorporating MS in the DR-scrubbing setup, as the two methods produce mean
baselines with a relative difference within the measurement uncertainty of the instrument. Therefore, we combine the
measurements from DR and DR+MS measurements. This subset of consecutive 4.5-min averaged baselines is shown
in Fig. 6. DNPH-scrubbed air resulted in normally distributed baseline measurements with a mean and standard
deviation of -0.76 ± 0.19 ppb. The baseline distribution resulting from DR- and DR+MS-scrubbed air is also normally
distributed with a larger mean and but lower standard deviation of -0.29 ± 0.15 ppb. Ambient HCHO concentrations
were often low enough at night in the summer and throughout the day in late autumn/winter to produce negative
differential concentrations in the range of –2.0 to –1.0 ppb HCHO for ambient air.

The humidity range reported by the G2307 when sampling air scrubbed by a DNPH-coated cartridge is 0.3–3% $H_2O$,
which is on average two orders of magnitude larger and exhibits significantly greater variability than the corresponding
range of 0.01–0.15% $H_2O$ (mean of 0.05% $H_2O$) when sampling through DR or DR+MS. Previous studies have noted
that derivatization of hydrazine to hydrazone, which is the reaction that functionally captures HCHO in the cartridge,
is impeded at low relative humidities (<15 % RH) or does not proceed if in completely dry conditions (Wisthaler et



al., 2008; Uchiyama et al., 2009). This is a limitation on the scrubbing method as deployment in completely arid locations would then significantly reduce the hydrazone yield, allowing uncaptured HCHO to pass through. However, converting the instrument-reported humidity to RH using the atmospheric conditions of the trailer that housed the G2307 reveals a minimum of 18.6 % RH. Though the trailer is climate-controlled and removes most water from its intake, conditions were still sufficiently humid year-round for optimal derivatization. In absolute terms, ambient air must have a composition of at least 0.25 % $H_2O$. In conjunction with results from Glowania et al. (2021), measurements are then made solely in a regime that is 15 times less sensitive to changes in humidity.

As mentioned before, Whitehill et al. (2018) noted an inverse correlation between HCHO and instrument-reported humidity at concentrations in the range of 0.5-3% $H_2O$, which is above the 0.2% $H_2O$ threshold found in Glowania et al. (2021). When comparing ambient measurements from the G2307 with a co-located Aerodyne TILDAS HCHO analyzer (Aerodyne Research, 2022), the G2307's reported HCHO decreased by approximately 0.8 ppb as instrument-reported humidity increased from 1-2.8% $H_2O$. Glowania et al. (2021) note the algorithm employed in their instrument, which was released following the Whitehill et al. (2018) intercomparison, was designed as an improvement to the fitting procedure at low humidity. Whitehill et al. (2018) employed DR in their instrument setup, which would have reduced instrument-reported humidity to a range comparable to that observed in this work and thus well below the 0.2% $H_2O$ threshold. While higher $H_2O$ contributed to low biases while sampling ambient HCHO, baseline measurements were possibly biased high by as much as 1.5 ppb HCHO. Combined, these effects can fully explain the low bias observed in both their measurements as well as in Furdyna (2020).

Given the narrow range of instrument-reported humidity when sampling through DR observed in this work, we cannot reliably determine a regression for baseline offsets at ≤0.2 % $H_2O$ for comparison with Glowania et al. (2021). However, as our instrument employs the same spectral fitting procedure and measurements derived via DR-scrubbing are biased low, we expect similar behaviour is exhibited. During our G2307's deployment, both ambient and DNPH-scrubbed measurements were taken over a similar range of $H_2O$ concentrations to those observed in Whitehill et al. (2018) and Glowania et al. (2021). Though our instrument was housed in a climate-controlled unit where DNPH-cartridges sampled indoor air, instrument-reported $H_2O$ concentrations for zeroed measurements increased over time as the cartridge became more saturated. To assess if a similar relationship occurs above 0.3% $H_2O$, baseline averages for the complete DNPH-scrubbed baseline dataset were plotted against instrument-reported humidity, but no notable correlation was determined. These results give more confidence to DNPH-based measurements as it indicates that DR does not remove all HCHO from incoming sample and has a more limited range of suitable ambient conditions.



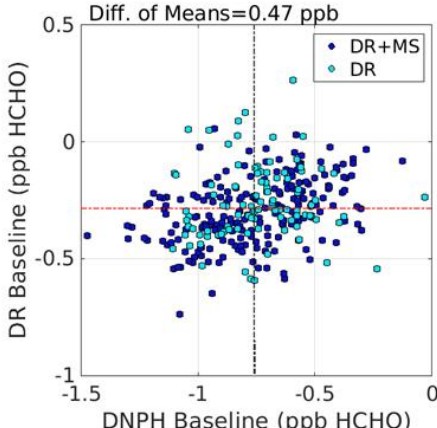

**Figure 6 – Picarro G2307 baselines determined using the DR, DR+MS, or DNPH scrubbing methods. Each data point represents a consecutive DNPH and DR baseline measurement, averaged over 4.5 min according to the instruments data scheme in Sect. 2.2. The difference of means is calculated by subtracting the mean of the DNPH measurements from the mean of DR and DR+MS measurements.**

## 4.3 Impact of scrubber choice – heated HO and DNPH for Aeris MIRA

Following a similar procedure to the previous section, we used scrubbed-air measurements from the Aeris Pico to assess differences in HCHO by comparing removal via heated HO to removal via DNPH-scrubbing. The $H_2O$ produced from HO oxidation of organic compounds was well in excess of the 2000 ppm $H_2O$ threshold of the instrument, making this method a suitable candidate for use with the Aeris instruments. The Pico alternated between these two scrubbers at 40 s intervals over a 30 min period.

DNPH-scrubbed baselines again exhibited a normal distribution, this time centered around a mean of -13.63 ± 0.54 ppb. HO-scrubbed baselines exhibit a normal distribution with a higher mean of -12.92 ± 0.34 ppb resulting in a difference of means of 0.71 ppb, again indicating less efficient HCHO removal. While HO-scrubbed baselines exhibit comparably better precision, the Pico's Allen-Werle curve showed that at typical time resolutions, DNPH-scrubbing results in precisions high enough for ambient monitoring. Both methods produce sufficiently high humidity for spectral fitting, with the Pico reporting a range of $2 – 2.4 \times 10^4$ ppm $H_2O$ while sampling through HO and a range of $1.8 – 2.5 \times 10^4$ ppm $H_2O$ while sampling through DNPH-coated cartridges. Throughout the Aeris instruments' deployments, only a few outlier days fell below the 2000 ppm $H_2O$ threshold. DNPH-coated cartridges when used on the Aeris instruments typically last 5-8 days depending on ambient conditions and atmospheric chemical composition. Given the zeroing sequence used for these measurements, a cartridge then has a corresponding breakthrough time of 17 – 27 h. Additional regressions comparing residual size from spectral fits, ambient HCHO concentration, and baseline values to instrument reported $H_2O$ concentration were performed to determine potential instrument humidity



dependencies but no meaningful correlations were established. This demonstrates that seasonal variability in ambient conditions does not significantly impact Aeris measurements and that DNPH-coated cartridges are suitable through all seasons at this location.

## 5 Instrument intercomparisons

We compare observations from the Aeris Pico's co-location periods with the other two continuous HCHO monitors, as the Pico's portability allows for easier transfer between the two sites. A York regression of 20 min averaged data is used for all comparisons in Fig. 7a and Fig. 7b (York et al., 2004). This technique uses instrument measurement uncertainties, described previously in Sect. 2, to increase the dependence of the line of best fit on more precise measurements. Instrument baselines are measured using the DNPH-scrubbing method.


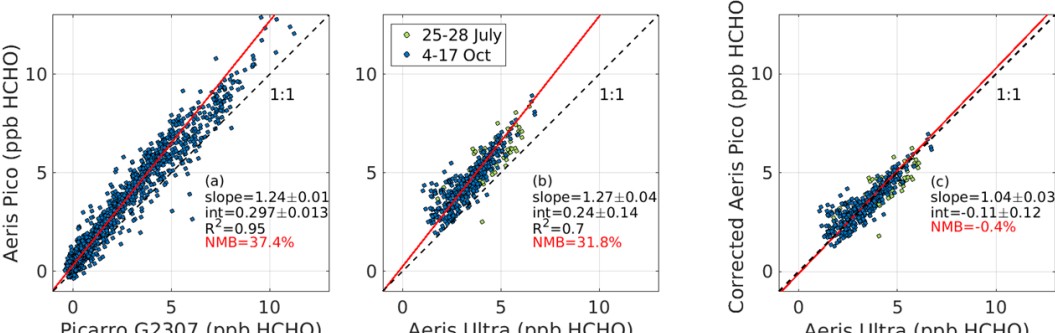

**Figure 7 – Aeris Pico ambient time series correlations with the two other HCHO monitors assessed in this work. Data are averaged to a 20 min time resolution and are from the Pico's respective co-location periods with either (a) the Ultra at GT or (b) the G2307 at SDK. (c) shows the Pico's observations corrected using the Pico v G2307 regression.**


The Pico's observations are consistently biased high, evidenced by producing two regressions with slopes of 1.24 and 1.27 and intercepts of 0.30 and 0.24, which implies good agreement between the G2307 and the Ultra. To correct for the Pico's bias, we use the Pico v. G2307 regression as its correlation is higher and the number and range of observations are larger. Results for the corrected data are shown in Fig. 7c. The resulting Pico v. Ultra NMB is reduced

to -0.4 %, the resulting slope close to unity with a value of 1.04, and the intercept reduced to -0.11 ppb. While the Pico may have drifted from factory calibration before deployment, the consistent comparisons in July and October suggest that the sensitivity was stable over multiple months.

Finally, we compare measurements from the Picarro G2307 with those from co-located 8 h DNPH samples (Fig. 8).

Concentrations from the G2307 are averaged to the same 8 h sampling windows after converting from ppb to $\mu g\ m^{-3}$ using ambient temperature and pressure data from the SDK site. Observations are well correlated ($R^2 = 0.82$), but the



Picarro G2307 has an offset of 1 µg m$^{-3}$ HCHO and a slope of 1.47. The positive bias is in contrast with the previous intercomparisons conducted by Whitehill et al. (2018) and Furdyna (2020) who, again, both found that the G2307 measurements were consistently lower than the DNPH-samples by 1-2 ppb HCHO. Given that measurements from

these two studies were taken at different field sites, during two different seasons, and over an instrument-reported humidity range like that observed in our data, it is anticipated that this low bias results largely from the instrument itself and its setup. As stated before, discrepancies between this work and these previous studies are attributed to the spectral fitting algorithm used in the previous intercomparisons and the choice of scrubber.

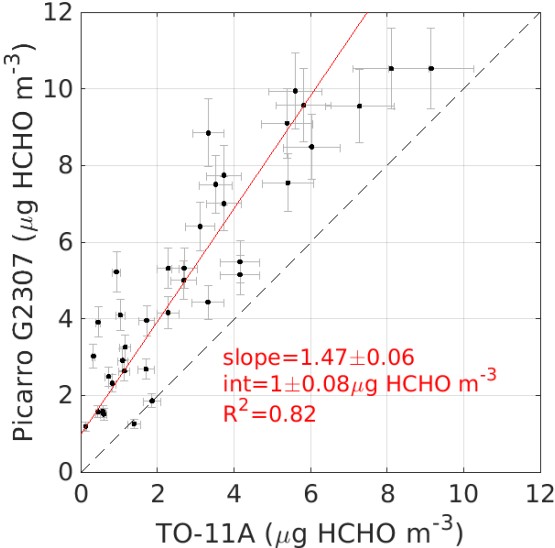

**Figure 8 – 8 h TO-11A DNPH samples correlated to Picarro G2307 observations both taken at the SDK site from June through July 2022. Concentrations reported by the G2307 have been integrated over corresponding 8 h collection periods. Error bars represent the 12.25 % and 10 % error associated with the DNPH and G2307 measurements, respectively.**


Furthermore, previous urban field studies have demonstrated that the HPLC analysis of DNPH-derivatized HCHO produces observations that are biased low relative to observations from continuous HCHO monitors (Hak et al., 2005; Dunne et al., 2018). Hak et al. (2005) intercompared a suite of continuous HCHO monitoring instruments and 2 h integrated DNPH-samples, finding the latter's measurements to be biased low by as much as 25 %. Of note to this

work are their regressions between DNPH-samples and HCHO concentrations determined via a research-grade Hantzsch fluorometric monitor, which had sub-unity slopes of 0.64 and 0.83. From Glowania et al. (2021), a comparison between their G2307 and a Hantzsch monitor utilizing a similar setup found close agreement between the two instruments (slope of 1.08, $R^2 = 0.97$), thus we would anticipate a slope greater than unity for our G2307 v DNPH regression. In this context, measurements from the G2307 show marked improvement, exhibiting high precision at





time resolutions down to 5 min and requiring significantly less operational input. This conclusion extends to the Aeris instruments as well given their close measurement agreement with the G2307.

## 6. Suitability for long-term deployment

To demonstrate whether these continuous HCHO monitors capture the urban HCHO gradient, we plot time series

from both field sites from Aug. 2022 – Jan. 2023 (Fig. 9) and quantify the HCHO concentration gradient that arises between Atlanta's urban core (GT) and a less industrialized, rural-urban area (SDK). Gaps in data typically result from downtime due to scrubber exhaustion or instrument maintenance. The Aeris instruments overall have less available data due to more frequent and intense scrubber usage, valve failures, and spectral fitting failures that could not self-correct. Over this 6 mo period, the Pico was stationed at both field sites with data after 18 Oct 2022 unavailable

as it was dedicated to other experiments.

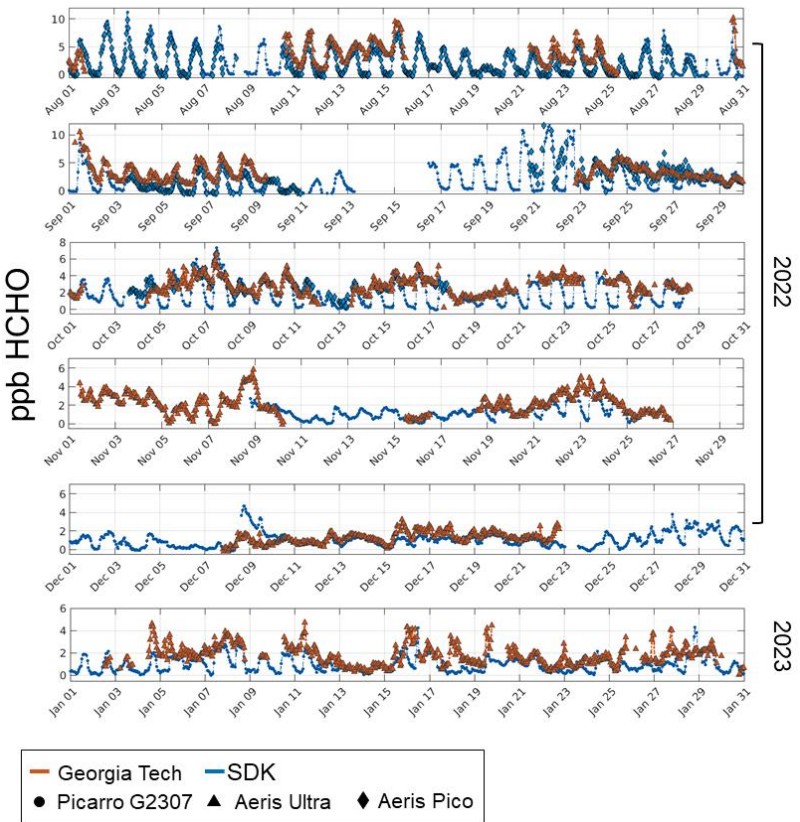



**Figure 9 – 1 h averaged HCHO time series from Picarro G2307, Aeris Ultra, and corrected Aeris Pico from Aug. 2022 through Jan 2023. Observations at GT show less defined diurnal amplitudes than the SDK site and are on average higher despite the time of year. Aeris Pico data is unavailable past 18 Oct. 2022 as it was dedicated to other experiments.**


In August, both sites reached their daily maximums around 13:00 LT with values of 6.37 ppb HCHO at GT and 5.11 ppb HCHO at SDK. On average, HCHO concentrations were 1.77 ppb higher than the SDK site, with monthly maximums of 9.85 ppb at GT and 8.38 ppb at SDK both occurring on 15 Aug. Measurements at Georgia Tech

generally show a less defined diurnal pattern with a night-time minimum concentration around 1.5 ppb. Given that the SDK site is located in a less urbanized area and immediately surrounded by trees with perennial leaves, this trend matches results found in Wang et al. (2022), who noted that cities with higher levels of biogenic VOCs exhibited larger HCHO diurnal amplitudes. As such, we expect that the influence of isoprene chemistry on HCHO production is stronger at SDK. This poor diurnal definition at GT further degrades as the year progresses into colder months,

whereas the SDK site maintains clear patterns with night-time minimum values of 0.08 ppb in August and 0.05 ppb in the winter. This consistent night-time threshold at GT could result from a combination of an anthropogenic, primary HCHO emission source local to the city and from stagnant atmospheric conditions leading to localized changes in night-time surface layer mixing heights. Fig. 9 spans over a long enough time to capture the extremes in ambient conditions of the metropolitan area, showing that the observed differences in HCHO concentration between the two

sites are well within the measurement capabilities of the G2307 and the Aeris instruments.

Our data also allows for a snapshot comparison with previous measurements at both sites to look at changes in HCHO concentrations. While HCHO data (collected via the TO-11A methodology) is officially available up to April 2022 at the SDK PAMS site (at the time of submission), the only prior ground-based campaign to measure HCHO via a

continuous monitor in the Atlanta urban core was the 1999 Atlanta Supersite Project (Solomon et al., 2003), where a Hantzsch fluorometric monitor was deployed for the month of August (Dasgupta et al., 2005). HCHO observations taken in the urban core are used to calculate an August diel cycle for their respective years after being converted from ppb HCHO to µg m$^{-3}$ using corresponding ambient temperature and pressure measurements. To extend this analysis to both sites, we employ the PAMS HCHO data taken at SDK in July 1999 (AQS, 1999) to compare with the July

2022 data previously used in the G2307 v TO-11A regression (data beyond July is unavailable at the time of submission). In 1999, DNPH samples were collected every 3 h starting at 06:00 and ending at 18:00 LT, meaning they do not capture a complete diurnal profile. As stated previously, samples are now collected every 8 h over a 24 h period starting at 04:00 LT. As such, a 6 h average of the 1999 observations (12-18:00 LT) are compared with the 2022 8 h average (12-20:00 LT) with the results shown in Fig. 10.






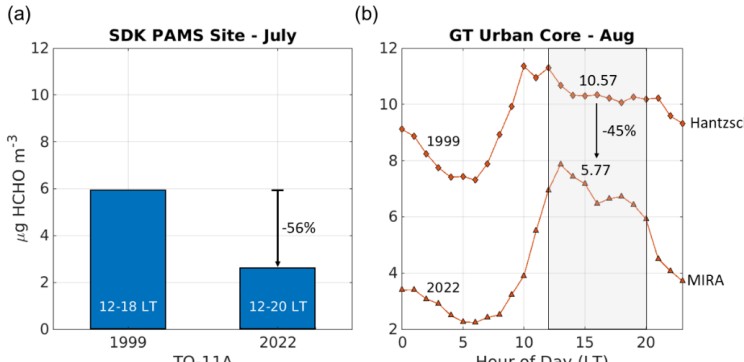

**Figure 10 – HCHO concentrations from 1999 and 2022. (a) shows a 56% decrease from July 1999 to 2022 in the midday**
**average (12-20:00 LT) of PAMS measurements taken at SDK and (b) shows a 45% decrease for the same averaging window**
**in August in Atlanta' urban core.**

An average of the August HCHO observations over the 12-20:00 LT window show that concentrations at GT have
reduced by 45% since 1999 despite the increasing urbanization of the city over the last two decades. The average
decrease in daily minimums and maximums at GT are 61.6% and 32.7%, respectively. Dasgupta et al. (2005) state the
possible influence of nearby HCHO emission sources, but this remains nonetheless a considerable decrease in
concentrations. Correspondingly, an even greater midday decrease of 56% at the SDK site is determined. Minimum
and maximum values can't be calculated for SDK as the 1999 data doesn't span the entire night-time. Given that the
SDK values are derived via the TO-11A method, we anticipate the absolute ambient HCHO values to be
underreported.

Continuous measurements provide the benefit of comprehensive time series, meaning local chemical trends of HCHO
can be more clearly related to time-dependent atmospheric conditions. In the urban core, maximum HCHO
concentrations always occur in the daytime and minimums in the night-time, with the maximum relative difference
being less than the minimum. OH oxidation of isoprene is one of the dominant sources of HCHO in urban
environments that have sufficiently high $NO_x$ concentrations, with the southeast having comparably higher biogenic
influences on its atmospheric chemistry than the rest of the country (Travis et al., 2016). As significant reductions in
U.S. $NO_x$ emissions have been observed over the decades (Duncan et al., 2016), urban, daytime HCHO production is
then expected to decrease. As OH is largely a daytime oxidant, night-time decreases in HCHO are more likely
attributable to reductions in direct emissions of both HCHO as well as its anthropogenic VOC precursors. These results
ultimately show that the fast and accurate observations from these HCHO monitors mean that deployment across
multiple cities is both feasible and has the potential for greater insights into the complex chemistry of urban HCHO.





## 7 Conclusions


We used long-term ambient datasets from three commercially new in-situ HCHO monitors to quantify instrument performance and to compare observations with measurements produced from co-located monitors employing the EPA TO-11A methodology. These continuous monitors offer a potential advantage to the TO-11A measurements given their high precision and comparably finer time resolution. However, previous measurements exhibited humidity

dependencies, produced significantly lower concentrations, and showed non-negligible variability in HCHO concentration dependent on zeroing method. Additionally, all three instruments utilize absorption-based spectroscopy and require frequent zeroing via HCHO scrubbers to account for baseline drift with each method presenting its own set of practical considerations. To determine an optimal field setup, we assessed how measurement quality changed with usage of four common scrubbing methods: DNPH-coated cartridges, DR and DR+MS adsorption columns, and

a thermally activated HO column.

DR, DR+MS, and HO-scrubbed baselines were compared to those resulting from DNPH-coated cartridges. Heated HO performed poorest, exhibiting the largest mean differential baseline value (0.71 ppb HCHO). The DR and DR+MS scrubbing methods performed better, but still led to baseline values with a mean differences of 0.47 ppb HCHO when

compared to DNPH-scrubbed baselines. These results indicate inefficient removal of HCHO from ambient air and an inability to operate in ambient conditions with low HCHO concentrations, which occurred nightly during spring and summer, and constantly during winter and autumn. Additionally, the G2307 has shown high baseline offsets at concentrations $\leq 0.2\ \%\ H_2O$ and an inverse correlation with HCHO at $> 0.2\ \%\ H_2O$ which together can explain the low biases in HCHO observations seen in previous EPA-conducted intercomparison efforts. This effect can be

mitigated by the choice of scrubber, namely using DNPH-coated cartridges over DR. We note that DNPH-scrubbing has a minor limitation when used in conditions with low humidity ($< 15\ \%$ RH or $0.25\ \%\ H_2O$) but found that our field site had perennially sufficient humidity for optimal HCHO capturing.

As DNPH-coated cartridges performed best, we employed this method for instrument intercomparisons. We found

that the Picarro G2307 meets manufacturer-stated performance metrics (0.07 ppb precision for a 5 min average) and exceeds this value over a 20 min integration time (0.055 ppb). This instrument also exhibits the comparatively lowest baseline drift (1.1 ppb over the course of a week). The Aeris Ultra and Pico reach a precision of 0.055 ppb and 0.08 ppb, respectively, for the same 20 min integration window, which exceeds the previously reported precision of 0.42 ppb. The Aeris Ultra experiences approximately 4 ppb of drift and the Pico nearly 20 ppb over a 1-week period. We

attribute the Pico's poorer performance to the comparatively lesser thermal insulation in the unit. We find that the Aeris Pico also had a consistently high bias of 31.7 % and 38.4 % when compared to Aeris Ultra and Picarro G2307 measurements, respectively. Correcting its observations using the Pico v G2307 regression led to all measurements agreeing within 5%. Our comparison of the G2307 with co-located TO-11A observations show that the DNPH-sampling measurements were biased significantly low and had a significant offset of 1 ppb. This contrasts with

previous EPA intercomparisons wherein G2307 observations were generally lower by up to 2 ppb. We attribute this



discrepancy to 2 factors: (1) older versions of the spectral fitting algorithm used and (2) the use of DR scrubbers in both EPA studies. Furthermore, comparison of our G2307 v. TO-11A regression to those determined in previous studies shows that the low bias in TO-11A measurements is expected, lending greater confidence to the G2307's observations.


Ultimately, we determine that these instruments offer a clear advantage to the existing TO-11A methodology, providing high precision and accuracy at fast time resolutions. Using time series that span from Aug. 2022 through Jan. 2023 at two fields sites separated by 15 km, we demonstrated that these instruments capture differences in the HCHO gradient in the Atlanta metro area over a wide range of ambient conditions which encompass summer and

wintertime seasonal extremes. Comparison with historical HCHO measurements revealed a relative decrease in ambient HCHO of approximately 50 % at both sites since 1999. The performance of these instruments showcases the feasibility of deploying across multiple cities and the potential insights to be gained thereafter.

**Competing Interests.**

The authors declare that they have no conflicts of interest.

**Acknowledgements.**

The authors kindly thank Jaime Gore and DeAnna Oser from the Georgia Environmental Protection Division for
furnishing data used in this work and all their input thereafter.

**Financial Support.**

This research has been supported by the National Aeronautics and Space Administration (grant no. 80NSSC21K0944).

**Data availability.**

All data shown in this work as well as any forthcoming are available at: https://doi.org/10.5281/zenodo.7682263





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
