# Peer review of "Evaluation of Aeris MIRA, Picarro CRDS G2307, and DNPHbased sampling for long-term formaldehyde monitoring efforts"

_EGUsphere, 2023_

## Author Response (AR1)

We thank the reviewers for their insightful comments and suggestions. The original general comments are below in black, our responses follow in blue, and the revised manuscript text in red.

We preface our response by noting three significant additions to our prior analysis.

(1) A humidity-dependent sensitivity was determined for the Picarro G2307 instrument. The procedure for quantifying and accounting for the humidity impact is discussed in Sect. 2.1.3:

**2.1.3 Humidity-dependence**

Two trials were performed to quantify the impact of humidity on G2307 measurements. HCHO-free air was provided by either a zero-air (ZA) generator (Tofwerk) with DR column (trial 1) or an ultra ZA cylinder (trial 2). A portion of the ZA stream was humidified by using a bubbler containing Milli-Q water. The fraction of ZA humidified was varied using a mass flow controller such that measured H2O concentrations ranged from 0.05-1.7%.

Fig. 1 shows the reported HCHO concentrations in HCHO-free air as a function of measured % $H_2O$. As reported in Glowania et al. (2021), data fell into two linear regimes with a demarcation at 0.2 % $H_2O$. Data were averaged to 5 minutes and each regime fitted using a York regression (York et al., 2004) with standard deviations of the measurements used as uncertainty. We find significantly smaller slopes (lower $H_2O$ influence) than Glowania et al. (2021), indicating that humidity-dependencies may be instrument-specific. The HCHO offset is defined in Eqn. 1:

$$[HCHO]_{offset} = \begin{cases} (-5.67 \pm 0.47) * [H_2O] + (0.13 \pm 0.02), \% \, \text{H2O} \, < \, 0.2 \\ (-0.40 \pm 0.02) * [H_2O] - (0.01 \pm 0.02), \% \, \text{H2O} \, \geq \, 0.2 \end{cases} \quad (1)$$

where [HCHO]$_{offset}$ (ppb) accounts for the HCHO signal lost at some % $H_2O$ and [$H_2O$] is the corresponding instrument-reported % $H_2O$ mole fraction.

Depending on the instrument zeroing method, ambient and baseline humidities may be very different. These differences could lead to signfcant biases in reported HCHO differential measurements. For example, Fig. 1 suggests the use of a dessicant such as DR, for sampling ambient air at 1% $H_2O$ would generate a bias of -0.4 ppb if the humitidy dependence is not corrected. We emphasize the importance of experimentally determining a correction factor for humidity-effects before deployment.

[Figure]

**Figure 1 – Picarro G2307 HCHO concentrations as a function of measured H₂O concentrations. Regressions for the two H₂O spectral fitting regimes are plotted alongside the slopes from Glowania et al. (2021). Error bars are the standard deviation in instrument baseline or % H₂O for each 5 min averaged point.**

(2) Dynamic dilution calibrations of Picarro G2307 were performed, as described in section 2.1.6.

**2.1.6 Instrument calibration**

[revised manuscript text omitted]

The identification of humidity-dependent sensitivities and recalibration of the instruments have been taken into account for all data. As a result, the prior conclusions of the original manuscript have been altered:

(1) We now find that all zeroing methods except for heated hopcalite are effective for instrument zeroing.
(2) Sensitivity for all HCHO monitors remained stable (relative change within uncertainty) throughout their respective deployment periods, and after calibrating, all monitors agreed within 13%.
(3) The Picarro G2307 v TO-11A regression results in an even greater difference between methods (Fig. 9).
(4) The comparison with 1999 SOS data now shows a smaller decrease in urban daytime HCHO concentrations from 1999 to present day (Fig. 11).

**Referee #1:**

**General Comments:**

I had several general concerns with this paper. The first has to do with how the calibrations were performed:

1) The Picarro was calibrated at 100x to 1000x higher concentrations than the range measured in the field, and was also calibrated in a different matrix gas (N2 vs Air). The authors did point out that they assume linearity, but they do not provide strong evidence for linearity from single-point calibrations, and certainly not linearity over 3 orders of magnitude.

> We thank the referee for bringing these points to our attention. Not mentioned in the original manuscript was the correction factor used to convert the instrument reported HCHO concentration in an $N_2$ matrix to an air matrix (a factor of 1.0625). This detail has now been included in Sect. 2.1.6 (above):
>
> **Lines 231-232:** "The single-point measured concentration was determined as the instrument-reported concentration mutiplied by an $N_2$/air matrix conversion factor of 1.0625 (Bent, 2023)."
>
> Furthermore, we have performed an additional, multi-point calibration for the Picarro that spans a range of 0 – 40 ppb HCHO in an air matrix. A description of all the Picarro calibrations has been added in Sect. 2.1.6. (above).
>
> These calibrations demonstrate the consistent sensitivity of the Picarro G2307 over the course of its deployment, with resulting regressions having slopes from 2021-2023 within 8% of unity and within 10% of one another.

2) The Aeris instruments either weren't calibrated (Ultra) or were post-corrected based on comparison with the Picarro (Pico). Therefore, the slope and intercept in Figure 7 are essentially a test of how good the Aeris calibration is. Given the 15+ months from manufacturing to use, the strong comparison is impressive but perhaps not very useful.

> To address this concern, we've performed a dynamic dilution calibration as well as a second round of standard addition calibrations for both Aeris units, discussed in Sect. 2.2.5 (above). The results indicate negligible change in their sensitivities throughout deployment.
>
> We have also performed a second round of intercomparisons with all three instruments co-located, now shown in Sect. 4.2.1. (above). We found that after applying various calibrations that all instruments agreed within 13% of each other. Calibrations are discussed in sections 2.1.6 and 2.2.5.

3) Given the importance of appropriate calibrations for making these measurements, I recommend the authors spend additional time discussing recommendations for how to calibrate the instruments. Even suggestions based on experience (if presented as such) would be better than completely ignoring the issue.

> We've incorporated this suggestion by adding recommendations for calibrating the Aeris and Picarro units throughout sections 2.1.6 and 2.2.5 (above), which detail each instruments' calibrations via multi-point, standard dilutions in zero air or standard additions.

**From Sect. 2.1.6:**

**Lines 227-233:** "Single-point and dynamic dilution calibrations were conducted at the beginning, middle, and end of the G2307's deployment. Single-point calibrations were performed by flowing a concentrated standard (either Apel Riemer: 1015 ppb ± 5%, Airgas: 1031 ppb ± 10%, or Airgas: 1044 ± 10%) through a silonert-coated stainless steel (SS) regulator and directly into the instrument. This configuration avoids interaction between the calibration gas and stainless steel surfaces, thereby reducing passivation times to sub-hour lengths. However, this technique relies on the assumption that observations are linear from 0-1 ppm HCHO. The single-point measured concentration was determined as the instrument-reported concentration mutiplied by an $N_2$/air matrix conversion factor of 1.0625 (Bent, 2023)."

**From Sect. 2.2.5:**

**Lines 312-314:** "In Sept 2023, both Aeris instruments were calibrated using dilutions of a HCHO gas standard (either Apel Riemer: 1015 ppb ± 5%, or Airgas: 1044 ppb ± 10%) with humidified ultra-ZA. The configurations for humidifying air and diluting the gas standard were as described in sections 2.1.3 and 2.1.6."

The second major concern is with the suggestion that the Picarro G2307 be run with a DNPH scrubber for regular zeros.

4) I would make sure to mention that doing this would likely (and probably should) void the manufacturer's warranty and/or any maintenance or service agreements and could also reduce the lifetime of the instrument.

We appreciate the point made here and contacted a representative from Picarro, Inc. to help address it. They stated through personal correspondence that use of DNPH-coated cartridges does not invalidate the user warranty. They are unsure as to whether prolonged use leads to a shortened lifetime of the G2307. To address this uncertainty, we have included a note in Sect. 2.1.5 that mirror degradation from use of DNPH-coated cartridges could be examined more closely (lines 225-226):

**2.1.5 Impact of scrubber choice – DNPH, DR, and DR+MS**

Before comparing scrubbers, we first examine the HCHO-removal efficiency of DNPH compared to a ZA generator. We find instrument baselines were on average 14 ppt larger than those measured using a ZA generator. This difference was consistent whether sampling the indoor conditions or ambient air. This difference is not statistically significant given the instrument precision and accuracy determined later in Sect. 3. We note DNPH initially off-gases material that produces spectral interferences that subside after a "burn-in" period of ~2 hrs. It's possible that off-gassing material could have negative effects on instrument performance if used long-term (e.g., mirror degradation). These impacts were not seen in our study and would require further investigation.

The impact of DR and DR+MS on the Picarro G2307's baseline was then assessed using ambient measurements taken from the consecutive sampling of DNPH and DR/DR+MS in the ambient sequencer schedule. We combine the DR and DR+MS measurements as we find the two methods produce baselines with a relative difference that is within instrument measurement uncertainty. The 4.5-min averaged baselines are shown in Fig. 2. Both scrubbing methods produced normally distributed baseline measurements with means and standard deviations of -0.39 ± 0.14 ppb

(DNPH) and -0.38 ± 0.15 ppb (DR/DR+MS), and an average absolute difference of <0.03 ppb HCHO. This difference is finer than the 5 min precision of the instrument and demonstrates a comparable performance between the two scrubbings methods.

[Figure]

**Figure 2 – Picarro G2307 baselines determined using the DR, DR+MS, or DNPH scrubbing methods. Each data point represents a consecutive, 4.5-min averaged DNPH and DR baseline measurement.**

Previous studies have noted that derivatization of hydrazine to hydrazone, which is the reaction that functionally captures HCHO in the DNPH-coated cartridge, is slowed or stopped at RH < 15 % (Wisthaler et al., 2008; Uchiyama et al., 2009).

Few days throughout the G2307's deployment fell below this threshold, and RH (converted from instrument-reported % $H_2O$ using indoor conditions) was always ≥ 25 %. While low RH likely did not affect our measurements, we note this is a limitation on DNPH as deployment in arid locations could hamper performance whereas DR/DR+MS would operate unaffected.

Ho et al. (2014) found that high temperatures (>22 °C) and RH (>50%) led to DNPH-HPLC analysis underestimating ambient HCHO by 35-80%. This could inflate instrument baselines as summer 2022 in Atlanta regularly exceeded these values, with DNPH-derived baselines in Fig. 2 having RH values in the range of 7-87%. As DR-baselines are determined using desiccated air and the average baseline difference with DNPH is within instrument precision, we conclude measurements are not significantly affected at high RH. These results lead us to conclude that either DR/DR+MS or DNPH usage with the G2307 is advisable so long as humidity corrections are applied."

5) The LpDNPH S10L cartridges are made by coating silica gel with 2,4-dinitrophenylhydrazine in an acidified solvent solution. As a result, the LpDNPH S10L cartridges will emit small amounts of solvents (possibly acetonitrile or methanol), acid gases (e.g., HCl or H2SO4), and possibly volatilized byproducts of the DNPH reactions. It is not immediately clear what impact those might have on the cell or mirrors in the G2307, or whether any of those compounds might contribute to spectral interferences in the formaldehyde retrievals in the 5626 cm⁻¹ region.

To the point of the referee, we have added a section discussing short term effects of DNPH-cartridge offgassing on the Picarro G2307's HCHO absorption spectrum in Sect. 2.1.5.

**Lines 191-194:** "We note DNPH initially off-gases material that produces spectral interferences that subside after a "burn-in" period of ~2 hrs. It's possible that off-gassing material could have negative effects on instrument performance if used long-term (e.g., mirror degradation). These impacts were not seen in our study and would require further investigation."

To provide further detail, we've included Fig. R1 below, which shows the Picarro G2307's instrument-reported HCHO peak absorption. A comparison of the G2307's baseline when sampling through a DNPH-coated cartridge (following the burn-in period) or a zero-air generator reveals that absorption at the HCHO peak is effectively equivalent as regressions between the two methods have negligible intercepts and a slope of 1.02, leading us to conclude that there are no short-term spectral interferences.

[Figure]

**Figure R1 – HCHO peak absorption at a wavelength of 1.78e-4 cm as reported by the Picarro G2307 when sampling through either a DNPH-coated cartridge or a zero-air generator. This period is binned via instrument-reported % $H_2O$ when sampling through DNPH.**

6) This should also be considered when discussing the ~0.5 ppbv offset difference between the DR / DR+MS baseline and the DNPH baseline for the Picarro instrument. Although the DNPH cartridge makes sense for doing humidity-matched baseline corrections, it appears (Figure 3) that the Picarro is plumbed such that the zeros are performed from indoor air passing through the DNPH scrubber. In this configuration the instrument loses the ability to perform humidity-matched zeros, which seems to negate the major reason for using DNPH over a different scrubber (e.g., humidified zero air or just dry zero air). A zero-air cylinder would last longer than a DNPH cartridge, not have the potential concerns about acid / solvent off-gassing, and could last indefinitely with an appropriate zero air generator.

We thank the reviewer for these remarks. Our new humidity-dependent data processing scheme for the Picarro G2307 measurements has significantly altered our appraisals of DR/DR+MS and DNPH-coated cartridges. Details of the impetus for and derivation of this correction are provided in Sect. 2.1.3 of the revised manuscript, found at the beginning of this document.

Importantly, the application of this correction also removes the need to match humidities between ambient and zero measurements. After applying the humidity-correction to the data, the difference in DR/DR+MS, and DNPH-derived instrument baselines is now <30 ppt.

Furthermore, we conducted an additional experiment comparing baselines from a Tofwerk zero-air generator, and two DNPH-coated cartridges – one sampling indoors and the other outdoors. Mean relative differences of 5 min integrated baselines between the HCHO-scrubbers are within 12 %, corresponding to a mean absolute difference of 14 ppt HCHO. This is now stated in Sect. 2.1.5

**Lines 188-291:** "Before comparing scrubbers, we first examine the HCHO-removal efficiency of DNPH compared to a ZA generator. We find instrument baselines were on average 14 ppt larger than those measured using a ZA generator. This difference was consistent whether sampling the indoor conditions or ambient air. This difference is not statistically significant given the instrument precision and accuracy determined later in Sect. 3."

Lastly, while cylinders or zero-air generators do offer the benefits noted by the reviewer, we opt for HCHO scrubbing systems as they are comparatively easier to transport, require minimal setup and space, and have no power consumption needs. Again, we emphasize that the comparisons between a zero-air generator and DNPH, as well as DNPH and DR/DR+MS, showed all zeroing methods perform adequately.

Nowhere in the manuscript is the issue of the actual accuracy of the Aeris or Picarro instruments addressed.

The instrument accuracy for the Picarro G2307 is now stated in Sect. 2.1.6:

**Lines 242-243:** "We determine the uncertainty in ambient measurements to be 10 % per the uncertainty associated with the standards used for calibration."

The accuracy in the Aeris units now stated in Sect. 2.2.5:

**Lines 343-345:** "Correspondingly, a 14 % relative uncertainty from propagating the measurement uncertainties of the G2307 and Ultra in quadrature. A 0.3 ppb offset is added per the Pico/FILIF comparison in Shutter et al. (2019), which falls within the range of calibration offsets seen in this work."

7) Given the large discrepancy between the Picarro G2307 and the TO-11A, there are clearly accuracy issues with one or both methods. In addition, the discrepancies observed here are larger than those observed in many other studies comparing DNPH to spectroscopic measurements.

We thank the referee for this insightful comment. To this point, there was an accuracy issue with the Picarro G2307 in that observations were not corrected for humidity. However, applying the correction exacerbates the discrepancy in question. We have provided additional analysis attempting to determine the reason for this discrepancy, discussed in Sect. 4.2.2:

**4.2.2 Picarro G2307 and TO-11A DNPH comparison**

Fig. 9 compares G2307 observations from June-Aug. 2022 with those from co-located TO-11A measurements. 1 min integrated G2307 concentrations are averaged to the 8 h TO-11A sampling window. We find moderate correlation (r = 0.62) and a -58 % NMB of TO-11A observations relative to the G2307 (slope = 0.38 ± 0.02). Previous studies have demonstrated DNPH-based observations being up to 25 % lower relative to continuous HCHO observations (Hak et al., 2005; Dunne et al., 2018). Hak et al. (2005) determined slopes in the range of 0.64-0.83 when comparing DNPH-HPLC and Hantzsch fluorometric measurements. A comparison of Hantzsch and G2307 observations in Glowania et al. (2021) produced a slope of 1.08.

While a low bias is not unusual for TO-11A measurements, the magnitude of the discrepancy presented here is larger than prior studies. We find 8 h G2307 observations are well correlated (|r| > 0.7) with temperature, RH, and $O_3$, which are expected to either drive ambient HCHO or reflect its secondary chemistry. In contrast, TO-11A observations had weak correlations with these same variables, attaining a maximum r of 0.44 with $O_3$ and |r| ≤ 0.20 for all others. TO-11A observations did not correlate notably with $NO_2$ which would be expected to bias reported HCHO concentrations high (Herrington and Hays, 2012). Noted in Sect. 2.1.5, summertime in Atlanta exhibits high RH and temperatures, which can lead DNPH measurements to underestimate ambient HCHO by 35-80% (Ho et al., 2014). While we are unable to provide a definite reason for this significant discrepancy, the accuracy and stability shown through the G2307's calibrations as well as its agreement with the Aeris units (with independently verified accuracies) lend confidence to its measurements.

[Figure]

**Figure 9 – 8 h TO-11A DNPH observations compared to Picarro G2307 observations at the SDK site from June through August 2022. Error bars represent the 10% uncertainty associated with the TO-11A and G2307 measurements.**

8) I also have concerns about the 0.5 ppbv difference in baseline using the DNPH versus DR and whether that could contribute to the 1 ug/m3 intercept in the Picarro vs DNPH comparison. Nowhere do the authors make a convincing argument that the DNPH zero is a "true zero" versus the DR or other zero / scrubber mechanism.

We refer back to the end of comment (6) wherein DNPH-scrubbing performed comparably to sampling from a zero-air generator. We believe this is satisfactory in demonstrating that DNPH-coated cartridges are a "true zero" (and, by extension, DR/DR+MS columns).

**Section 5:**

If I understand this correctly, the Aeris Ultra essentially used the manufacturer's calibration, the Picarro was calibrated at 1 ppmv in $N_2$, and the Aeris Pico was corrected based on the Picarro. Therefore, the comparability of the Aeris Pico, Aeris Ultra, and Picarro G2307 (especially the slope and intercept) seem to be a test of how comparable the Ultra calibration is to the Picarro calibration. Essentially, this section seems to be a test of the Aeris Technologies calibration more than actual instrument-versus-instrument comparability. It might be more interesting to look more closely at:

9) When the disagreement between methods occurred (e.g., is there any pattern that could suggest a reason for the disagreement beyond just random noise)?

10) How large was the "scatter" around the regression line – was it in line with what would be expected from the precision / Allan variance measurements reported earlier, or is there additional uncertainty factors from measuring in ambient air? If the latter, could you quantify how large that additional uncertainty is and suggest possible causes?

> We thank the referee for bringing up these topics in comments (9) and (10). With the addition of the second intercomparison, the multi-point calibration for the Picarro, and the second round of standard addition calibrations, we have individually verified each instrument's sensitivity. Precisions for the three instruments have been reassessed in lieu of referee comments, now discussed in Sect. 3:

**Lines 373-426:**

**3. Instrument precision and baseline drift**

The precisions of the three analyzers were characterized in two ways. First, the instruments' inlets were overflowed using a ZA source for 24 h and precision was calculated via an Allan-Werle curve, as in prior instrument characterization studies (Shutter et al., 2019; Glowania et al., 2021). Results are shown as the solid lines in Fig. 4. The G2307 achieves precisions of 0.09 ppb, 0.05 ppb, and 0.03 ppb for integration times of 5, 20, and 60 minutes. This performance is similar to the 5 min 0.06 ppb precision reported by the manufacturer and results determined in Glowania et al. (2021). The Ultra achieves precisions of 0.20 ppb, 0.20 ppb, and 0.28 ppb for the same periods. The best precision achieved by the Pico is 0.66 ppb at a 30 s integration time. At longer integration times, fluctuations in concentrations reported by the Pico instrument can be attributed to thermal instability. Internal instrument temperatures varied by ±0.3-0.4 °C over the course of 7 h and were well-correlated (r > 0.85) with the instrument baseline. Resultingly, precisions past 40 s integration times quickly became unsuitable for ambient monitoring. During deployment, the Pico's internal temperature was more stable compared to the ZA tests performed in the laboratory. When using 30 s zeroing periods from the Pico's ambient time series, a precision of 0.40 ppb HCHO is determined, which is comparable to that of the Ultra for the same integration window.

[Figure]

**Figure 4 – Allan-Werle curves for a) Picarro G2307 b) Aeris Ultra and c) Aeris Pico instruments. Uncorrected precisions (solid lines) are calculated without accounting for baseline variation, whereas corrected precisions (dashed lines) use the same baseline-characterization method used to process ambient data.**

As Allan variance is not meant to address systematic errors like temperature effects, we developed a modified, or corrected, Allan-Werle curve that better characterizes the precision of ambient measurements. Still sampling ZA, we replicated the sampling sequences and data processing methods used for ambient measurements (i.e. the 1 Hz data is drift-corrected by averaging and subtracting out each zeroing period). We then treated the 1 Hz data measured on the "ambient" inlet as contiguous. Results are shown as dashed lines in Fig. 4. For the Picarro G2307 (Fig. 4a), there is no change in precision using this method, as the baseline is relatively constant in this period. Both Aeris units benefited significantly from this correction, reaching 40 min precisions of 0.140 ppb and 0.154 ppb for the Ultra and Pico, respectively. The Pico's modified precision is within 15 ppt of the 40 min precision of 0.14 ppb observed in Shutter et al. (2019). The corrected Aeris Allan-Werle curves trend similarly to the G2307's, achieving lower precisions with longer integration times. These results indicate that the ambient sampling sequences used for each instrument are sufficient to account for the influence of any physical instrument-variables on the baseline. As the precision of the ambient measurements (which are calculated differentially) is impacted by both the precision of the ambient and zero baselines, the modified Allan-Werle curves do not account for the precision of the zero measurement. In our ambient dataset, we are limited to a 30 s integration time per the sampling sequence of the Aeris units. The 30 s Allan deviation while sampling through DNPH in our ambient dataset is 0.45 ppb for the Ultra. For the Pico, observations prior to Aug. 2023 have a precision of 0.41 ppb and 0.66 ppb otherwise. This is taken as the true precision of the ambient dataset. Longer zeroing times may achieve higher precision in the dashed lines of Fig. 4 if the baseline has sufficiently low drift through the sampling period.

The scatter around each line of best fit as well as additional factors of uncertainty are now discussed in Sect. 4.2.1:

**Lines 498-503:** "The scatter around the lines of best fit is primarily owed to the low precision of the Aeris ambient measurements, which is determined by the 30 s zeroing intervals. There are occasional periods of large deviations from the lines of best-fit. These periods typically lasted multiple hours, suggesting accuracy (rather than precision) is the cause of the deviations. Specifically, on four separate occasions the Aeris instruments both measured 10-15 ppb HCHO while the Picarro observations remained at ~10 ppb. Reasons underlying this behavior could not be traced to measured instrument parameters or ambient variables."

**General Suggestions:**

PLEASE use consistent units when talking about formaldehyde concentrations. Choose either ppbv or µg·m⁻³ and don't keep switching back and forth in different parts of the text and different figures. Conversion from one unit set to the other is straightforward, but it's very difficult as written to compare concentration ranges in different sections because the units are not consistent.

We apologize for any confusion introduced by converting units. Figures comparing TO-11A and Picarro G2307 measurements, as well as comparing HCHO concentrations between 1999 and 2022, now use data provided by the EPA's Air Quality System (AQS) which are in units of ppb. These changes are reflected in sections 4.2.2 (above) and throughout sect. 5.

**5. Suitability for long-term deployment**

[revised manuscript text omitted]

**Specific Comments and Suggestions:**

Line 36: Recommend "Because HCHO photolysis / oxidation is a source of …" instead of just "HCHO is a source of"

> **Lines 32-34:** "Because photolysis of HCHO is a source of $HO_x$ radicals, HCHO loss can further propagate oxidative chemistry (Tonnesen and Dennis, 2000; Lin et al., 2012; Valin et al., 2016; Wolfe et al., 2019; Yang et al., 2021)."

Line 44: Recommend "the standard EPA approach" rather than "EPA-standard"

> **Lines 41-43:** "Since 1990, the standard EPA approach for HCHO measurements is collection on 2,4-dinitrophenylhydrazine (DNPH) coated cartridges followed by offline derivative detection via high performance liquid chromatography (HPLC), known as the TO-11A method (U.S. Environmental Protection Agency, 1999)."

Lines 45-46: Please cite the 1999 version of EPA Method TO-11A. You cite Riggin, 1984, which is before the TO-11A "Method" existed. (https://www.epa.gov/sites/default/files/2019-11/documents/to-11ar.pdf)

> **Lines 41-43:** "Since 1990, the standard EPA approach for HCHO measurements is collection on 2,4-dinitrophenylhydrazine (DNPH) coated cartridges followed by offline derivative detection via high performance liquid chromatography (HPLC), known as the TO-11A method (U.S. Environmental Protection Agency, 1999)."

Line 46: I believe it should be "Sample collection and analysis are" instead of "is"

> **Lines 43-44:** "Sample collection and analysis are resource and labor intensive with measurements typically reported over sampling times that are on the order of hours."

Line 47: "…long sampling times" is very ambiguous. TO-11A is generally used for 1 hour to 24 hour sampling in ambient air – it is not effective if the time is too short (not enough HCHO collected) or too long (you get breakthrough).

> We've altered the text to be more specific regarding the sampling times of the TO-11A method.

> **Lines 43-46:** "Sample collection and analysis are resource and labor intensive with measurements typically reported over sampling times that are on the order of hours. EPA TO-11A measurements in the PAMS and NATTS are 8 or 24 h integrated samples collected every three or six days, respectively."

Line 47-48: "EPA Method TO-11A measurements in the PAMS and NATTS networks are 8 or 24 h …" Technically TO-11A measurements can be any length – you are specifically talking about current PAMS and NATTS required sampling frequency / duration.

> **Lines 45-46:** "EPA TO-11A measurements in the PAMS and NATTS are 8 or 24 h integrated samples collected every three or six days, respectively."

Line 51: "the method…" here seems to refer to the "previous approaches" in the sentence prior rather than "Method TO-11A" (which I believe is the intended target for "the method").

> **Lines 47-53:** "Previous approaches have used modelled diel cycles or satellite-based observations in combination with the TO-11A method to infer ground-based diel cycles (Zhu et al., 2017a; Zhu et al., 2017b; Wang et al., 2022). However, this DNPH method of capturing HCHO has known interferences from $NO_2$ and $O_3$ (Karst et al., 1993; Achatz et al., 1999; Tang et al., 2004), can have a variable collection efficiency (CE) dependent on relative humidity (RH) or is not suitable for sampling at <15 % RH (Wisthaler et al., 2008; Uchiyama et al., 2009; Ho et al., 2014), and has had mixed results in comparison to research-grade observations (Hak et al., 2005; Wisthaler et al., 2008; Dunne et al., 2018), making the accuracy of these inferred diel cycles difficult to determine."

Line 51: Perhaps mention "the DNPH method…" because Method TO-11A specifically addresses the O3 interference (which was actually a large impetus for publication of TO-11A versus staying with TO-11).

> **Lines 49-53:** "However, this DNPH method of capturing HCHO has known interferences from $NO_2$ and $O_3$ (Karst et al., 1993; Achatz et al., 1999; Tang et al., 2004), can have a variable collection efficiency (CE) dependent on relative humidity (RH) or is not suitable for sampling at <15 % RH (Wisthaler et al., 2008; Uchiyama et al., 2009; Ho et al., 2014), and has had mixed results in comparison to research-grade observations (Hak et al., 2005; Wisthaler et al., 2008; Dunne et al., 2018), making the accuracy of these inferred diel cycles difficult to determine."

Lines 57 – 61: There are a number of datasets with about 1 month or longer of continuous spectroscopic formaldehyde measurements at ground level, generally using TDLAS. See, for example, Coggon et al. (2021) (https://doi.org/10.1073/pnas.2026653118, Figure S20) or Spinei et al. (2018) (https://doi.org/10.5194/amt-11-4943-2018, Figure 3).

> We've now included Spinei et al. (2018) and Coggon et al. (2021) references in this passage but have removed further references to MAX-DOAS campaigns as this technique does not provide in-situ measurements.

**Lines 53-57:** "While other studies have demonstrated the feasibility for continuous measurements via various spectroscopy-based methods (Yokelson et al., 1999; Cardenas et al., 2000; Dasgupta et al., 2005; Hak et al., 2005; Spinei et al., 2018; St Clair et al., 2019; Dugheri et al., 2021), the number of multi-month, ground-based, continuous, in-situ HCHO measurements is limited to a handful of studies, all of which a proton-transfer-reaction mass spectrometer (Warneke et al., 2013; Hansen et al., 2014; Coggon et al., 2021)."

Line 63: "A more suitable long-term HCHO monitoring instrument…" – more suitable than what? And suitable for what purpose? I recommend rewriting this entire sentence – it's a bit confusing as written.

We've altered the manuscript to hopefully better convey our argument for what constitutes the criteria for an effective, long-term ambient HCHO monitor.

**Lines 59-61:** "A HCHO monitoring instrument more suitable for long-term deployment would reduce manual labor and provide continuous observations, experience little or correctable drift in instrument baseline and sensitivity, and have low uncertainty and sufficient precision at typical ambient concentrations."

Line 77: "relies on the HDO line" – which HDO line? Either specify a line or say "a HDO line"

**Lines 75-76:** "The Aeris MIRA technique relies on a HDO line (located at 2931.8413 cm$^{-1}$) for spectral referencing."

Line 88-89: "This updated algorithm…" – are you referring to the algorithm used in Glowania et al. (2021) or the post-Glowania algorithm update to resolve the issues reported in Glowania et al. (2021).

We apologize for the confusion. The updated algorithm – released Sep. 2019 – is the algorithm used in Glowania et al. (2021). This update came after the Whitehill et al. (2018) study and is also used by our Picarro G2307 unit.

**Lines 67-68:** "Glowania et al. (2021) is the only peer-reviewed work to employ a G2307 using the current spectral fitting algorithm (version 1.6.015), which updates the procedure for fitting at low-humidity."

**Lines 83-85:** "Glowania et al. (2021) found that variable humidity can decrease reported HCHO concentrations by as much as 1.75 ppb with the most significant offsets at $\leq 0.2\%$ $H_2O$ where the $H_2O$ spectral feature is not clearly observed."

Line 91: Technically, the Picarro G2307 does not "rely" on periodic instrument baseline zeroing. Once calibrated, it should be stable for months without needing to zero. Regular zeroing is recommended for the highest (sub 1 ppbv) precision (e.g., minimize baseline drift).

We've altered the text so as not to imply that observations cannot be produced unless the G2307's baseline is sampled.

**Lines 87-91:** "Both Picarro and Aeris instruments periodically sample HCHO-free air to determine an instrument baseline. Several scrubbers are capable of removing HCHO – the most common of which are DNPH-coated cartridges (DNPH), heated catalytic hydrocarbon scrubbers like oxides of copper and manganese (hopcalite, HO), calcium sulfate (Drierite, DR), and molecular sieves (MS) (Herndon et al., 2007; Cazorla et al., 2015; Pei et al., 2015; Shutter et al., 2019; St Clair et al., 2019; Fried et al., 2020)."

Line 92 – 94: A commonly used scrubber for HCHO-free air is a heated catalytic hydrocarbon scrubber. This is often used in cases where humidity-matched zeros are necessary. See, e.g., Herndon et al. (2007) (https://doi.org/10.1029/2006JD007600). I believe it is also used by Fried et al. on various aircraft studies. This is also used in commercial zero air generator systems to produce HCHO-free air.

> We've included the provided Herndon et al. (2007) reference and as well as Fried et al. (2020) to convey the wide usage of heated catalytic hydrocarbon scrubbers. This study also now makes use of a Tofwerk zero air generator, which uses a platinum catalyst heated to 400 °C. Our work specifically tested heated hopcalite (oxides of copper and manganese) as an inexpensive alternative with an operating principle similar to zero air generators used in the aforementioned references. The resulting lines can be found in the comment above this one.

Lines 150 – 160: It's odd for the authors to calibrate the HCHO at > 1 ppmv but make most of their measurements in the 1 – 10 ppbv range. This is a 3 order of magnitude difference between the calibrated range and the measured range. Linearity across 3+ orders of magnitude is a major assumption, especially given the potential influence of peak shape on formaldehyde retrievals at different concentrations over a 3 order of magnitude range. Because these are single-point calibrations, the authors do not even test the linearity of the instrument across any range.
In addition, the calibrations were done in an N2 bath gas, whereas the zero and measurements were done with a N2/O2 mix (air). In an ideal situation, calibration matrix would match the measurement matrix as closely as possible, especially considering the high potential for matrix effects in a high-reflectivity cell with > 1 km effective pathlength. Given the 50% discrepancies observed versus DNPH, I recommend some of the assumptions made during the calibration be reconsidered (or at least discussed more thoroughly).

> We believe this point is fully addressed in our response to comment (1).

Lines 150 – 160: The concentration of HCHO in reference gas cylinders typically decrease over time at a pseudo-linear rate. It would be helpful to know when the gas cylinders were certified by Apel-Riemer / Airgas versus when they were used to perform the calibration checks.

> We believe this point is now passively addressed in Sect. 2.1.6. To further expound – five standard dilution calibrations were conducted for the Picarro G2307 from 2021-2023. Throughout this period, three different HCHO standards were used. The Apel-Riemer cylinder (1015 ppb ± 5%, certified May 2019) had the longest time between certification and calibration – being used to calibrate G2307 in Oct. 2021 and Jun. 2022. Two Airgas cylinders followed, denoted AG1 (1031 ppb ± 10%, certified Aug. 2022) and AG2 (1044 ppb ± 10%, certified Aug. 2023). AG1 was used for calibration in Sep. 2022 and June 2023. AG2 was used in Sep. 2023. The resulting calibration slopes were within 8% of unity, 10% of each other, had negligible intercepts, and demonstrated high linearity ($R^2 > 0.99$) up to a ppm range. We believe this indicates little degradation had occurred between certification and calibration for any cylinder.

Line 166: The manufacturer's literature / spec sheets describe a 13 m pathlength for the instruments. I recommend you check whether 1.3 m or 13 m is the correct pathlength.

> We thank the author for catching this discrepancy and have adjusted the text to use the correct value for effective path length.

> **Lines 250-251:** "Air is pulled at a rate of 0.45 – 0.75 SLPM into a folded Herriott detection cell, which achieves a path length of 13 m."

Line 178 / 179: Figure 2 is a map. I believe the authors intend to refer to Fig 3a / Fig 3b in these lines. Figures should also, in general, be added to the manuscript in the order they are referred to in the text, which would make Fig 3 the first figure (Fig 1).

We thank the author for catching this discrepancy. This figure is now Fig. 7 in Sect. 4.1.1. and all references in the manuscript have been updated accordingly.

**4.1.1 South DeKalb**

The location of the South Dekalb (SDK) PAMS is shown in Fig. 6. The site is located approximately 12 mi southeast of the university campus in a less industrialized area with comparatively greater tree coverage. The G2307 was permanently stationed at SDK, with two intercomparisons performed during its deployment. First, the Aeris Pico was co-deployed from 28 July to 13 Sept 2022 according to the configuration shown in Fig. 7a. Then, the Aeris Ultra and Pico were co-deployed from 21-29 Aug. 2023 in their standard ambient configurations without sharing ambient lines. Instruments were housed in a climate-controlled trailer with an indoor temperature maintained at 21-23 °C. All tubing was 0.125 in ID (0.25 in OD) PTFE with 7.5 m extending from inside the trailer and up a mast, where the inlet was situated 5 m above the ground. The G2307 and Pico instruments had flow rates of 450 cm$^3$ min$^{-1}$, leading to a residence time of approximately 4 s when teed together, and 8 s when separate. The Ultra had a flow rate of 800 cm$^3$ min$^{-1}$ with a residence time of 5 s. 1μm particulate filters (PFs) in Savillex holders were used, and inlets were shielded by PTFE funnels covered with PTFE mesh. The indoor portion of the sampling lines were heated to 46 °C (≥1 °C above the cavity cell temperature of the instruments) to avoid condensation in the plumbing.

[Figure]

**Figure 6 – Locations of the two field sites in the Atlanta, GA area where the Aeris Ultra, Aeris Pico, and Picarro G2307 were deployed.**

The Aeris instruments' baselines were determined solely using DNPH while the G2307 sampled between DNPH, DR, or DR+MS. When scrubbing only with DR, air was passed through two adsorption columns

(length of 16 in, radius of 2 in) in series containing 0.5 kg of material each. For DR+MS, the column first in the series was replaced with the MS material. When the adsorption columns were exhausted, the scrubber bed was replaced with either new or regenerated material. DR was thermally regenerated according to the manufacturer instructions.

[Figure]

**Figure 7 – Configurations of instruments during their respective intercomparisons. (a) shows the teed setup used from 28 July – 13 Sep 2022 for the Aeris Pico and Picarro G2307. When not co-located, the G2307 has the same configuration without being teed to the Pico. (b) shows the setup used for the Aeris instruments while deployed at GT from 25 – 28 July and 4 – 17 Oct 2022. For each panel, "0" references HCHO-scrubbed air, "1" is ambient air, and "PF" is a particle filter."**

Lines 177 – 179: The "Scrubbing ambient air rather than indoor air…" part is confusing, since indoor air also has sufficient water vapor to maintain a laser line lock (and, in fact, Aeris markets their instruments for indoor air measurements of HCHO as well). Scrubbing ambient air will provide humidity-matched (or very close to humidity matched) background versus sample gases, whereas scrubbing indoor air would produce a near constant humidity for the zeros but a varying humidity for the sample gases.

> We apologize for any confusion. The intention of this line was to state that the Aeris units sampled outdoor air while the Picarro sampled indoor air. The text has been updated to hopefully clarify this and can be found in the above comment.

Line 179: It is not clear why authors choose to sample ambient air for 180 s and scrubbed air for 30 s, versus the scheduling used by Shutter et al. or recommended by the manufacturer.

> We thank the referee for this comment. For the revised manuscript, precision experiments using zero air were conducted for each HCHO monitor and have changed our appraisal of the instrument precision. The impetus behind our Aeris ambient scheduling sequence has been expounded on in Sect. 2.2.2:

**2.2.2 Instrument baseline**

The Aeris instruments have a two-inlet design allowing for determination of instrument baseline throughout the data collection process. We run the instruments in the "programmed" mode, which allows the user to select the duration of sampling through each inlet. The instruments also have a "differential" mode, which produces ambient HCHO concentrations using on-board baseline subtractions. The zero inlet was connected to either DNPH-coated cartridge or a heated HO (United Filtration) scrubber and teed with the ambient inlet to the main sampling line per the manufacturer's recommendation. We sample ambient air for

180 s and scrubbed air for 30 s. This sequence was determined through visual inspection of Aeris time series with the intention of minimizing DNPH-sampling time while maintaining sufficient precision for ambient monitoring. We found 180 s to be the longest length of time between zeroes that either unit achieved where the remained consistently stable. Both units were then set to the same schedule. This led to DNPH-coated cartridges lasting 5-8 days, corresponding to a breakthrough time of 17 – 27 h. Variability in breakthrough time is dependent on ambient conditions and atmospheric chemical composition.

Furthermore, the Allan-Werle curve for our Pico (solid line in Fig. 4c in the revised version) suggests that shorter zeroing periods (30 – 40 s) achieve the best precision. Both units were then set to this schedule, though we note the Ultra achieves its best precision at a 15 min integration time. This behavior is not believed to be general to these models as they do not reflect precision results from Shutter et al. (2019). We deem this ambient sequence to be sufficient for producing accurate measurements as evidenced by calibration and intercomparison results. Updated precisions are discussed in Sect. 3.

[Figure]

**Figure 4 – Allan-Werle curves for a) Picarro G2307 b.) Aeris Ultra and c) Aeris Pico instruments. Uncorrected precisions (solid lines) are calculated without accounting for baseline variation, whereas corrected precision (dashed lines) uses the same baseline-characterization method used to process ambient data.**

Line 216-217: Please provide the correct (1999) reference for EPA Method TO-11A.

This reference has been updated to refer to the 1999 TO-11A compendium.

Lines 219 – 221: What was the temperature of the heated inlet? What type of ozone denuder? I'm assuming based on the ATEC sampler that the ozone denuder was a KI-coated copper tube heated to 50 °C, but this is important to mention. Particularly as there are concerns with some types of ozone scrubbers in sampling formaldehyde (see, e.g., Ho et al. 2013, https://doi.org/10.4209/aaqr.2012.11.0313).

The requested information has now been included in Sect. 2.3.

**Lines 352-364: "Method TO-11A outlines in detail the EPA guidance on preparation of DNPH-coated cartridges and subsequent analysis through HPLC (U.S. EPA, 1999). Formaldehyde was measured using an ATEC Model 8000 Toxic Air Sampler over three consecutive eight-hour periods spanning a full 24 hours**

with samples collected every three days. Ambient air was drawn at a rate of 0.9 – 1.1 L/min through a KI-coated copper inlet heated to 50°C to remove $O_3$ before passing through a DNPH-coated cartridge (Supelco LpDNPH S10) which collected carbonyls in their non-volatile, carbonyl-hydrazone derivative form. The denuder is necessary as it minimizes potential $O_3$-related interferences in the resultant HPLC chromatograms (Vairavamurthy et al., 1992). At the end of the sampling period, the cartridges were capped and stored in a refrigeration unit at ≤ 4 °C until analysis. The cartridges were then eluted with 10 mL of acetonitrile (ACN) and the eluent analysed via a Waters HPLC-UV system with a temperature stabilized (25 ± 1°C), reversed phase C18-coated silica gel (1.7 μm particle size) column (Bridged ethyl hybrid, 2.1 mm x 50 mm ID) at 360 nm wavelength. The eluents used in the HPLC process were deionized $H_2O$ and ACN. The HPLC system was calibrated before each use with known concentrations of HCHO and field samples are analyzed in comparison to blank cartridges."

Lines 220 – 221: Please provide additional details about the cartridge (e.g., sorbent, DNPH loading, bed size, etc.). Supelco DNPH-C-18 is not a standard item – their standard DNPH cartridges use silica gel, not C-18. This seems to be an unusual / custom cartridge type.

> We've updated the product description in the text after reaching out the Georgia Environmental Protection Division for clarification. The corresponding text can be found in the comment above.

Lines 230 – 234: The 12% uncertainty seems very low. Estimated collection efficiencies are generally about 70 – 100% (see, e.g., https://projects.erg.com/conferences/ambientair/conf18/MacGregor_Ian_AirToxics_8-15_800_SalonE_POST_508.pdf) for HCHO. In addition, there is likely to be a negative bias in the DNPH because of breakthrough of HCHO, reverse derivatization reactions, degradation of hyrdozone, etc. In contrast, for a good chromatographic program (that resolves the NO2 artifacts) there should not be a positive bias in the measurements.

> We appreciate the reference provided by the referee and opt to evaluate the uncertainty in TO-11A measurements at 15%. We also note that we have switched the axes for each variable in Fig. 9 to provide a direct comparison with Hak et al. (2005). This is now reflected in the manuscript in sections 2.3 and 4.2.2.
>
> Using an uncertainty of 30% for the TO-11A measurements (which corresponds to the range of % mass recovered from a DNPH-cartridge across varying relative humidities from the provided reference) and using an uncertainty of 10% for the Picarro G2307 leads to a regression with a slope of (0.33 ± 0.02) and an intercept of (0.09 ± 0.05) ppb HCHO. This result does not change the conclusions of this work.

Figure 9: It is very challenging to see the difference between the three symbol types in this figure. I recommend choosing a different presentation if you want to clearly distinguish between the Picarro and two Aeris by the symbol style.

> We apologize for the lack of clarity in Fig. 9 (now Fig. 10 in the updated version). We've updated the color scheme, increased marker sizes, and converted the figure to SVG for higher resolution.

**Referee #2:**

**General Comments:**

Mouat et al performs an intercomparison between Aeris MIRA instruments (Pico and Ultra) and Picarro CRDS G2307 in order to evaluate how robust these instruments are for long-term HCHO monitoring. Comparison is also made to the standard EPA Method TO-11A. The authors compare instruments against each other to determine relative biases and offsets. Additionally, they also evaluate how well HCHO is removed by DNPH, Hopcalite (HO), Drierite (DR), and Drierite+Molecular Sieves (DR+MS), and found that DNPH was best suited for all instruments to generate HCHO-free air.

The authors also present a multi-month HCHO dataset taken at two locations in the Atlanta metro area and show that HCHO has decreased by ~50% over two decades by comparing recent measurements in 2022 with those in 1999. The level of analysis for this last section of the paper is appropriate for AMT.

In addition to usual revisions and minor technical corrections, this paper contains several major revisions and corrections that I describe more fully below which the authors need to address. Broadly speaking, the major revisions encompass how the authors are calculating instrument performance and how they compare the instruments against each other.

This paper fully falls within the scope of AMT and would be of interest to its readership. I would recommend publication but only after proper and full attention has been paid to the following revisions.

**Major Revisions:**

**Section 2: Aeris MIRA**

1) Line 179: "We sample ambient air for 180 s and scrubbed air for 30 s": The default settings of the Aeris MIRA instruments used to be sampling ambient air for 15 s followed by scrubbed air for 15 s, yet the authors have dramatically changed the timing of the cycle without providing substantial justification as to why their new cycle is better (other than maybe extending DNPH lifetime). The instrument's use case is for ambient monitoring (not fluxes), so more frequent sampling of the scrubbed-air line is fine.

> We thank the referee for these insightful comments. We first want to note standard Allan-Werle curves were determined per the referee's suggestion later in comment (7). For the revised manuscript, precision experiments using zero air were conducted for each HCHO monitor and have changed our appraisal of the instrument precision. This is now discussed in Sect. 3:

> **Lines 375-413: "**The precisions of the three analyzers were characterized in two ways. First, the instruments' inlets were overflowed using a ZA source for 24 h and precision was calculated via an Allan-Werle curve, as in prior instrument characterization studies (Shutter et al., 2019; Glowania et al., 2021). Results are shown as the solid lines in Fig. 4. The G2307 achieves precisions of 0.09 ppb, 0.05 ppb, and 0.03 ppb for integration times of 5, 20, and 60 minutes. This performance is similar to the 5 min 0.06 ppb precision reported by the manufacturer and results determined in Glowania et al. (2021). The Ultra achieves precisions of 0.20 ppb, 0.20 ppb, and 0.28 ppb for the same periods. The best precision achieved by the Pico is 0.66 ppb at a 30 s integration time. At longer integration times, fluctuations in concentrations reported by the Pico instrument can be attributed to thermal instability. Internal instrument temperatures varied by ±0.3-0.4 °C over the course of 7 h and were well-correlated (r > 0.85) with the instrument baseline. Resultingly, precisions past 40 s integration times quickly became unsuitable for ambient monitoring. During deployment, the Pico's internal temperature was more stable compared to the ZA tests performed in the laboratory. When using 30 s zeroing periods from the Pico's ambient time series, a

precision of 0.40 ppb HCHO is determined, which is comparable to that of the Ultra for the same integration window.

[Figure]

**Figure 4 – Allan-Werle curves for a) Picarro G2307 b) Aeris Ultra and c) Aeris Pico instruments. Uncorrected precisions (solid lines) are calculated without accounting for baseline variation, whereas corrected precisions (dashed lines) use the same baseline-characterization method used to process ambient data.**

As Allan variance is not meant to address systematic errors like temperature effects, we developed a modified, or corrected, Allan-Werle curve that better characterizes the precision of ambient measurements. Still sampling ZA, we replicated the sampling sequences and data processing methods used for ambient measurements (i.e. the 1 Hz data is drift-corrected by averaging and subtracting out each zeroing period). We then treated the 1 Hz data measured on the "ambient" inlet as contiguous. Results are shown as dashed lines in Fig. 4. For the Picarro G2307 (Fig. 4a), there is no change in precision using this method, as the baseline is relatively constant in this period. Both Aeris units benefited significantly from this correction, reaching 40 min precisions of 0.140 ppb and 0.154 ppb for the Ultra and Pico, respectively. The Pico's modified precision is within 15 ppt of the 40 min precision of 0.14 ppb observed in Shutter et al. (2019). The corrected Aeris Allan-Werle curves trend similarly to the G2307's, achieving lower precisions with longer integration times. These results indicate that the ambient sampling sequences used for each instrument are sufficient to account for the influence of any physical instrument-variables on the baseline. As the precision of the ambient measurements (which are calculated differentially) is impacted by both the precision of the ambient and zero baselines, the modified Allan-Werle curves do not account for the precision of the zero measurement. In our ambient dataset, we are limited to a 30 s integration time per the sampling sequence of the Aeris units. The 30 s Allan deviation while sampling through DNPH in our ambient dataset is 0.45 ppb for the Ultra. For the Pico, observations prior to Aug. 2023 have a precision of 0.41 ppb and 0.66 ppb otherwise. This is taken as the true precision of the ambient dataset. Longer zeroing times may achieve higher precision in the dashed lines of Fig. 4 if the baseline has sufficiently low drift through the sampling period."

The impetus behind our ambient scheduling sequence has been expounded on in Sect. 2.2.2:

**Lines 266-269:** "We sample ambient air for 180 s and scrubbed air for 30 s. This sequence was determined through visual inspection of Aeris time series with the intention of minimizing DNPH-sampling time while maintaining sufficient precision for ambient monitoring. We found 180 s to be the longest length of time between zeroes that either unit achieved where the remained consistently stable. Both units were then set to the same schedule."

The new cycle chosen by the authors looks problematic when I look at Fig 5 and see a multi-ppbv spread (6 ppbv HCHO for Ultra and nearly 20 ppbv for Pico) in the HCHO mixing ratio over several days when just sampling scrubbed air (that presumably has 0 ppbv HCHO). That spread is large enough to severely affect the accuracy of measured HCHO mixing ratios measured in the field.

To help address this concern, the authors need to explain how their new cycle timing was derived and show data to see how it performs against whatever is the default cycle timing used by Aeris.

2) Another zero should be performed and an Allen-Werle curve derived using the default cycle timing and plotted against their new cycle timing since this now falls within the scope of their paper.

Mentioned in comment (1), precision measurements using zero air were conducted for each monitor and have affected our results as discussed in Sect. 3 (the corresponding passage can be found above). To summarize, precisions for the three instruments have been reassessed in lieu of referee comments. Effectively, the ambient measurements for the Picarro G2307 are determined to be 0.09 ppb at a 5 min integration time. Precisions of the Aeris units' ambient measurements are limited to a 30 s integration time per their sampling sequences. For the Ultra, this is determined to be 0.45 ppb. The Pico began experiencing technical problems in Aug. 2023 when the second intercomparison was conducted. The resulting precision is determined to be 0.41 ppb before Aug. 2023, and 0.66 ppb afterwards. These values are also found to be the Pico's optimal precision. The G2307 does not have a minimum in its Allan-Werle curve within an hour of integrating and achieves a precision of 0.03 ppb at a 1 h integration time. Our Ultra's optimal precision is found to be 0.15 ppb at a 15 min integration time. While this answer does not directly perform what the referee has asked for, we believe having conducted these tests provides more elucidating information.

**Section 4.1: Instrument drift:**

The authors state multiple times that they drift corrected the data (e.g., Line 295, Line 305), but never specify:

3) how this drift correction was derived

We thank the reviewer for the opportunity to clarify what is meant by "drift". All instruments here require sampling through two lines: one sampling a HCHO-free air source ("baseline") and one sampling ambient air. The final ambient HCHO concentration is the difference between baseline and ambient concentrations. We use the term "drift" to refer to the rate-of-change of the baseline. This definition is stated explicitly in Sect. 2.1.2.

**2.1.2 Determining instrument baseline**

The G2307 measurements reported here differ from prior studies primarily in that we employed an external zeroing system. The system is equipped to sample from either DNPH (Supelco LpDNPH S10L), DR (Drierite, 8 mesh, >98% $CaSO_4$, <2% $CoCl_2$), or DR+MS (Sigma Aldrich Molecular Sieve, 0.3 nm zeolite beads) to regularly monitor and correct the instrument's baseline. Baseline is defined throughout this work as the signal reported by the instrument when sampling from a HCHO-free source and drift as the rate of change of the baseline. This setup was accomplished by connecting the G2307 inlet to a 3-way PFA solenoid valve which alternated between an ambient sampling line and a zeroing line. The zeroing line was

then connected to another 3-way PFA solenoid valve to which the scrubbers were attached. The instrument sampled from DR or DR+MS for 5 min of every hour. Every fourth hour, the instrument sampled for 5 min through DNPH either directly before or after sampling from DR. The relative order of DR/DNPH sampling was found to have no impact on reported instrument baselines.

4)  how generalizable is this drift correction (i.e., can it be applied to subsequent data collected by the instrument?). This should be rectified.

We believe this sampling sequence and resulting drift correction to be general to the HCHO monitoring instruments assessed in this work. Others with these instruments should be able to apply these sequences and see similar results, and all data collected in this work are subject to the respective drift-correction scheme of the instrument. In Sect. 3 of the revised manuscript (passage provided in comment (1)), we state what we determined to be the minimum rates at which each monitor needed to be zeroed.

5)  Additionally, the authors don't explicitly mention whether or not they drift-corrected their multi-month HCHO dataset. Should that be drift-corrected too?

All data shown has been fully corrected according to the observed instrument zeros, with the data processing schemes for each instrument described in sections 2.1.4 and 2.2.4.

**2.1.4 Data processing**

Averaged HCHO datasets at variable time resolutions (1 – 60 min) were created from the 1 Hz data using the following procedure: first, all 1 Hz data were corrected for humidity-effects by subtracting the [HCHO]$_{offset}$ from Eqn. 1. Observations made within 30 s of a valve change were removed and baseline measurements were then averaged to 4.5 min points and linearly interpolated to create an instrument background on the same time basis as ambient data. The interpolated baseline was subtracted from the 1 Hz ambient measurements. Baseline-corrected ambient data were averaged to the desired time resolution with any periods having <50 % data completeness discarded. Data was further screened to exclude points where scrubbers were exhausted and therefor unreliable.

**2.2.4 Data processing**

We generated temporally averaged datasets with variable time resolutions (1–60 min) using a data handling scheme like that of the Picarro G2307 observations. Zeroes are averaged to single points and interpolated to a 1 Hz resolution, subtracted from the 1 Hz ambient data, and ≥50 % data completeness is required for any averaging interval. We discard the first 5 s of measurements after a valve switch.

6)  At one point (Line 324), the fastest Pico drift rate was 1.67 ppbv HCHO h$^{-1}$, which is not acceptable for ambient monitoring. However, that drift rate seems highly variable (i.e., time-dependent) since it doesn't seem to always be changing at that rate as shown by Fig 5. Plus, the sign of the drift (either positive or negative) changes with time too. Stating that an instrument is drifting is a very serious claim since, when sampling ambient air, folks won't know if the change in HCHO mixing ratio is due to some underlying instrument baseline drift or a real change in mixing ratio.

We refer to comment (1) which contains Sect. 3 of the revised manuscript. The Allan-Werle characterization of precision for the Pico shows that its optimal value is achieved at integration times from 30-40 s and were around 0.66 ppb for Aug. 2023, and 0.41 ppb prior to. The dashed line in Fig. 4c indicates

that the chosen ambient sampling sequence sufficiently accounts for drift in the instrument at even its fastest rate. We believe the intercomparison and calibration results (sections 2.2.5 and 4.2.1, respectively, found at the top of the document) show that the Pico's measurements are accurate. We note that its high drift is likely specific to our unit given its technical problems during its precision measurement and since its Allan-Werle curve does not match well with results from Shutter et al. (2019).

Fig 4 and surrounding text:
I don't trust how this figure was derived since the whole purpose of an Allen-Werle curve is to help identify the integration time at which instrument drift becomes an issue. By correcting out instrument drift beforehand, the resultant Allen-Werle curve of course looks better, but it tells the reader nothing about the instrument's precision and long-term variability. Also, when I look at the large multi-ppbv changes in variability in Fig 5 for both the Ultra and Pico, I don't see how that corresponds to <100 pptv 1-sigma precisions for 20 min integration times and higher as indicated by Fig 4.

7) To help address this concern, the authors should derive Allen-Werle curves using *unaltered and uncorrected* HCHO mixing ratio data as reported directly from the instruments. Reporting this gives a better indication of instrument precision and provides better comparison back to what was reported in prior work. Based on my previous comment, I'm still left wondering whether this drift correction is generalizable since can it be consistently applied to ambient data when 0 ppbv HCHO air isn't being flowed into the instruments?

Per the referee's suggestion, we have followed through with determining unmodified Allan-Werle curves for each monitor. This is described in Sect. 3 (referenced in comment (1)).

Regarding how generalizable drift is: as the reviewer has noted, the reported baseline of the Aeris Pico can drift quickly. However, the baseline is sampled every 3 min, and can be accounted for through subtraction, with all ambient data corrected at a 3 min interval. The differential measurement (ambient minus scrubbed air), which is a measure of ambient HCHO, does not display any drift. No drift correction beyond the standard instrument baseline subtraction is applied to the long-term dataset and there is no generalizable rate of change in the baseline. Our analysis quantifies how fast this baseline can change in the field to provide a sense of how frequently baseline sampling is needed.

The importance of frequent baseline monitoring for drift correction cannot be overstated and is highlighted in the user manuals for the instruments shown in this work. As long as this drift is frequently monitored and corrected for, then the observed rates do not pose any issue for ambient sampling and data interpretation. We have attempted clarify and emphasize these points in all relevant sections.

8) If the magnitude of these new 1-sigma precisions is on the order of a few 100 pptv HCHO, how would this impact your analysis comparing HCHO removal from DR, DR+MS, HO, and DNPH? Were the differences in baselines large enough to not be encompassed by the instrument's LOD?

After applying a humidity-correction to the Picarro G2307 data, the resulting difference in baselines between DNPH and DR/DR+MS reduces to <0.03 ppb, which is encompassed both in the G2307's $3\sigma$ LOD and its precision for a 5 min integration time, indicating DR/DR+MS to be an effective HCHO-removal mechanism. This is discussed in Sect. 2.1.5.

**Lines 186-208:**

**2.1.5 Impact of scrubber choice – DNPH, DR, and DR+MS**

[revised manuscript text omitted]

**Section 2: Aeris MIRA**

Line 201:
"…significant deviation in sensitivity had occurred since its [Pico] last factory calibration": As denoted by the title, this is in part an instrument evaluation/comparison paper, so it's concerning when there's comparison of instrument accuracies against each other (Fig 7 and elsewhere), but there's also probable cause that the factory calibration for one of the instruments (i.e., Pico) was no longer even valid.

9) When reading about the high 30-40% bias in the Pico, readers will be confused (and misled) as to whether this is a real instrument bias, or if it is simply because the authors are using an instrument with an expired factory calibration. In other words, how generalizable is this bias since someone just reading the abstract (Line 19) or conclusion (Lines 566-577) may not realize there are very significant caveats to this quantitative statement?

10) To help address this concern, the authors could verify that the Aeris Pico has the correct calibration/sensitivity (just like they did for the G2307 before doing subsequent comparisons) and then compare against the other instruments. Alternatively, the Aeris Ultra factory calibration seemed valid (Line 210-211), so maybe just do the instrument comparison between the Ultra and G2307 (would need to fix Fig 7 appropriately and correct corresponding text throughout paper).

> We thank the referee for bringing up these points in comments (9) and (10). A second round of standard additions and a second intercomparison were conducted in Aug. – Sep. 2023. After applying standard addition calibrations to the Aeris units, all instruments agreed within 13%. We have adjusted the text in the manuscript to explicitly state this and believe the accuracy and sensitivity of each monitor has been verified. These results are detailed in sections 2.1.6, 2.2.5, and 4.2.1 (found at the top of this document).

**Minor Revisions, comments, and technical corrections:**

- Lines 14-15: "Baseline drifts over a 1-week period of ambient sampling of 1 ppb, 4 ppb, and 20 ppb": You're technically reporting the spread of values as opposed to a time-dependent trend, so please state that.

> We thank the reviewer for pointing this discrepancy out. This line has been removed in the updated manuscript.

- Lines 156: "measured concentrations were consistently 7% lower than expected": This also falls within the uncertainty reported for the Airgas HCHO gas cylinder standard (10%), so isn't it possible that the lower readings were simply due to the gas cylinder standard having lower HCHO than reported (which is definitely a possibility for HCHO gas cylinder standards)? Also, how long was HCHO allowed to flow through the MFC before calibrations were done? Generally speaking, several hours are necessary to passivate the MFC surfaces, and having a low flow is helpful to not waste calibration gas.

> We believe these points are now passively addressed in sections 2.1.6 and 2.2.5, found at the top of this document. HCHO was allowed to flow for several hours prior to any standard dilution or standard addition calibrations. Only after the instrument signal had plateaued would calibration commence.
>
> Five standard dilution calibrations were conducted for the Picarro G2307 from 2021-2023. Each Aeris instrument received two standard additions and one dynamic dilution calibration between 2022-2023. Throughout these periods, three different HCHO standards were used. Calibration slopes for the Picarro G2307 and Aeris Pico were in the range of 0.92-1.02. In lieu of both the referee's comment and the calibration results, we've removed the line in question.

- Line 150 - 160: Another concern when calibrating at such high mixing ratios (i.e., >1000 ppbv HCHO) to derive sensitivities is assuming linearity over three orders of magnitude (from 1 to 1000 ppbv HCHO). The authors state linearity is observed between 1-10 ppbv (Line 155), but if using a sensitivity derived at ~1000 ppbv HCHO, was it checked that linearity is observed between 10 and 1000 ppbv HCHO?

> We have now performed a dynamic dilution calibration for the Picarro G2307 to verify its linearity in both an ambient and a ppm HCHO range. This is described in Sect. 2.1.6, found at the top of this document.

- Line 277: "two mass flow controllers": Are the MFCs were being used as valves in this setup (either fully open or fully closed) to go between the DNPH and HO lines?

> The MFCs were being used as valves and this has now been explicitly stated in Sect. 2.2.3.
>
> **Lines 287-288:** "Two mass flow controllers were placed upstream of the scrubbers and used as valves."

- Line 291: "instrument baselines measured while sampling through a DNPH-coated cartridge": At this point in the text, the authors should explicitly say they are scrubbing HCHO from ambient air to derive their Allen-Werle curves.

> We believe this point is addressed in comment (1).

- Fig 5: Not sure what is going on during the evening of September 6 between all three instruments, but if some part of the sampling apparatus was changed at that time, then this data should be removed since the setup was altered.

> Errant data from 6 Sep. 2022 has been removed for the three HCHO monitors. This is reflected in what is now Fig. 5 of the revised manuscript, located in Sect. 3.

[Figure]

**Figure 5 – Instrument baseline time series for all three HCHO monitors plotted differentially to the first point in the time series. The Ultra and G2307, equipped with better thermal stabilization, show significantly less drift than the Pico.**

- Line 345: "differential concentrations in the range of -2.0 to -1.0 ppb HCHO for ambient air": Why are the magnitudes of ambient HCHO in air lower than when sampling scrubbed HCHO air? What are you trying to convey here?

> Initially, before the humidity correction was applied to the G2307's ambient observations, the baseline measured when sampling through DR was unimpacted by humidity, but ambient measurements were not. This means that ambient sampling was biased low and the unimpacted baseline was large enough to turn the differential (ambient) HCHO concentrations negative. With the humidity corrections, this is no longer an issue and has been deleted from the revised manuscript.

- Fig 7: The Aeris Ultra and Picarro G2307 should just be directly compared if the Pico's calibration isn't trustworthy.

> We believe this comment is addressed in comment (10).

- Fig 9: It was hard for me to see when and where the Pico was sampling. Also, please make this a vector graphic since it becomes blurry and hard to read when rasterized.

> Fig. 10 (previously Fig. 9) has been converted to an SVG format with an updated color scheme and increased marker sizes to hopefully improve clarity. We thank the reviewer for the suggested improvement.

[Figure]

**Figure 10 – 1 h averaged HCHO time series from Picarro G2307, Aeris Ultra, and Aeris Pico from Aug. 2022 through Jan 2023. Observations at GT show less defined diurnal amplitudes than the SDK site and are on average higher regardless of time of year. Aeris Pico data is sparse past 18 Oct. 2022 as it was periodically dedicated to other experiments.**

- Line 538-539: Two more reasons that should be mentioned include (1) these are online measurements and (2) don't require special handling/storage of samples or use of hazardous chemicals.

> We have incorporated the reviewer's feedback, now reflected in the updated manuscript.

> **Lines 603-605:** "These continuous monitors offer an advantage given that their measurements are online, have sufficient precision at finer time resolutions, and don't require special handling or storage of samples or hazardous chemical."

- Line 563: Authors are making an incorrect comparison between 3-sigma (i.e., LOD) and 1-sigma values.

> We thank the reviewer for catching this discrepancy and have updated the revised manuscript to correct this. Changes can now be found in Sect. 6.

> **Lines 622-624:** "The G2307, Ultra, and Pico achieved modified precisions of 0.05 ppb, 0.20 ppb, and 0.22 ppb for a 20 min integration time, respectively."

- Lines 16-18: The authors appear to be incorrectly reporting the 3-sigma LOD since the numbers cited are for 1-sigma based on what is presented in Fig 4.

> We again thank the reviewer for catching this discrepancy and have updated the revised manuscript to reflect this. Changes can be found in the above comment.

> **Lines 16-17** "We show that a modified precision estimate accounting for regular instrument zeroing results in values of 0.09 ppb, 0.20 ppb, and 0.22 ppb at a 20 min integration time for the G2307, Ultra, and Pico, respectively."

- Line 135: Do you mean Fig 3a (not Fig 2a)? This happens elsewhere in the manuscript too when I think you mean to reference Fig 3.

> This figure has been changed to Fig. 7 and references have been updated throughout the manuscript.

- Line 278: "the sample flow alternated between scrubbers in 40 s intervals": That's not what Fig 3c depicts.

> This figure has been changed to Fig. 7 and no longer includes the (c) component from its initial version. The text has been changed to hopefully better convey how the experimental setup operated. This change can be found in Sect. 2.2.3.

> **Lines 281-290:** "Ambient measurements of HCHO-scrubbed air from the Pico were used to assess the HCHO-removal efficiency of heated HO as compared to DNPH. The zeroing inlet on the Pico was teed to a DNPH-coated cartridge and a stainless-steel column (length of 8 in, radius of 0.75 in) containing 215 cm$^3$ of HO. The HO column was wrapped in high temperature heat tape, insulated in a fiberglass sleeve, and heated to 180 °C. Pei et al. (2015) found HO at this temperature achieved nearly 100 % HCHO removal and preserved the scrubber bed from $H_2O$ poisoning. A condensation trap and second PF were placed downstream of the HO column to protect the instrument against potential liquid $H_2O$ and particulate matter. Two mass flow controllers were placed upstream of the scrubbers and used as valves. The Pico sampled from its zeroing inlet while the incoming flow alternated between scrubbers in 40 s intervals. The first 10 s of data after every switch was removed to preclude any effects from valve-switching. This removal period was determined experimentally."

- Line 417: Replace "This technique" with "This linear regression"

    This line has been entirely altered in the revised manuscript.

    **Lines 155-157:** "Data were averaged to 5 minutes and each regime fitted using a York regression (York et al., 2004) with standard deviations of the measurements used as uncertainty."

- Line 536: Replace "long-term" with "multi-month"

    We've opted to use "year-long" in lieu of the additional data included for the standard dilution and addition calibrations, as well as the new intercomparison effort.

    **Lines 601-603:** "We used year-long ambient datasets from three commercially new in-situ HCHO monitors to quantify instrument performance and to compare observations with measurements produced from co-located monitors employing the EPA TO-11A methodology."

- Fig 3 caption: Be more explicit by defining abbreviations in caption and defining 1 (sample line) and 0 (zero line).

    This figure has been changed to Fig. 7 and the text has been changed to hopefully better convey how the experimental setups operated in Sect. 4.1.1.:

    **Lines 463-466:** "Figure 7 – Configurations of instruments during their respective intercomparisons. (a) shows the teed setup used from 28 July – 13 Sep 2022 for the Aeris Pico and Picarro G2307. When not co-located, the G2307 has the same configuration without being teed to the Pico. (b) shows the setup used for the Aeris instruments while deployed at GT from 25 – 28 July and 4 – 17 Oct 2022. For each panel, "0" references HCHO-scrubbed air, "1" is ambient air, and "PF" is a particle filter."

---

## Author Response (AR2)

**Report #1:**

**Specific Suggestions:**

Lines 95, 137 - Please specify type and pore size of the molecular sieve (e.g., 5 A pore size Zeolite molecular sieve)

> The suggested change is reflected in lines 136-138: "The system is equipped to sample from either DNPH (Supelco LpDNPH S10L), DR (Drierite, 8 mesh, >98% CaSO4, <2% CoCl2), or DR+MS (Sigma Aldrich molecular sieve, 3 A pore size zeolite beads to regularly monitor and correct the instrument's baseline."

Line 150 - Please specify how the Tofwerk zero air generator works. Is it a heated catalytic scrubber (temperature? Catalyst?), does it remove water vapor, etc.

> The referee's comment has been incorporated in lines 151-152: "The ZA generator uses a platinum catalyst heated to 400 °C and requires a DR column as it does not remove water vapor."

Line 150 - Ultra ZA - Please specify the purity of "Ultra ZA"

> We've included the manufacturer of the zero-air cylinder in the updated description. Product descriptions can be found on the manufacturer website.

> The description of the cylinder has been updated on lines 149-151: "Two trials were performed to quantify the impact of humidity on G2307 measurements. HCHO-free air was provided by either a zero-air (ZA) generator (Tofwerk) with DR column (trial 1) or an Airgas ultra zero grade cylinder (trial 2)."

Line 152 – "Milli-Q" is a brand that produces a range of water purities. Please specify the purity of water used (e.g., Type 1, 18.2 Mohm-cm resistivity, < 5 ppb TOC, etc.)

> The description of the high purity water used for the humidity-dependence calibration has been updated on lines 152-153: "A portion of the ZA stream was humidified by using a bubbler containing high purity water (Barnstead Genpure Pro,18.2 MOhm cm resistivity, <5 ppb total organic carbon)."

Line 153 - I'm assuming you mean volume mixing ratios when talking about H2O "concentrations" but you should probably specify this.

> We've specified units on lines 153-154: "The fraction of ZA humidified was varied using a mass flow controller such that the measured water vapor volume mixing ratios ranged from 0.05-1.7%."

Line 184 - I believe you mean "therefore" instead of "therefor"

> This grammatical error has been updated on lines 185-186: "Data was further screened to exclude points where scrubbers were exhausted and therefore unreliable."

Line 243 - Please specify how you come up with 10% uncertainty in ambient measurements. Given the +/- 10% uncertainty in the calibration gas (independent of precision, drift, etc.), it seems like the expanded uncertainty in ambient measurements should be larger than 10%.

> The relative uncertainty used for Picarro G2307 ambient measurements is expected to be dominated by the 10% uncertainty associated with our calibration standard and does not incorporate any additional terms related to precision or baseline corrections.

Line 355 - Define KI (potassium iodide?)

> We've incorporated the reviewer's suggestion in lines 358-360: "Ambient air was drawn at a rate of 0.9 – 1.1 L/min through a potassium iodide-coated copper inlet heated to 50°C to remove $O_3$ before passing through a DNPH-coated cartridge (Supelco LpDNPH S10) which collected carbonyls in their non-volatile, carbonyl-hydrazone derivative form."

Line 376 - Define "Allan-Werle curve" and / or cite the relevant literature (e.g., Werle et al...)

> A reference to the description of Allan variance has been added on lines 378-380: "First, the instruments' inlets were overflowed using a ZA source for 24 h and precision was calculated via an Allan-Werle curve (Allan, 1966), as in prior instrument characterization studies (Shutter et al., 2019; Glowania et al., 2021)."

Line 396 - Define "Allan variance" and / or cite the relevant literature.

> This comment is addressed in the one above.

Lines 423 - 425 - How do you get the 3 minute / 10 minute / hourly zeroing intervals from your measurements? Does this assume particular threshold values for accuracy, precision, or drift? It's not clear how you came up with these values.

> These recommended sampling times were determined through visual inspection as discussed in Sect. 2.2.2. We've updated the manuscript to clarify how we arrived at these sampling intervals.

> Lines 426-429: "From our observations, we determined that the Pico should be zeroed at least every 3 min and the Ultra every 10 min under typical indoor-deployment configurations as the instrument-reported HCHO signals do not consistently remain stable at longer intervals. For the G2307, observations of the instrument baseline drift obtained using DR suggest that hourly zeroing is sufficient."

Line 448 - Specify filter type (e.g., 1 micron pore size PTFE filters) and filter holder material (e.g., Savillex PFA filter holders).

> We have altered the manuscript to include these distinctions.

> Lines 452-453: "1μm PTFE particulate filters (PFs) in Savillex PTFE holders were used, and inlets were shielded by PTFE funnels covered with PTFE mesh."

> Lines 479-480: "As before, a 1μm PTFE PF in a Savillex PTFE filter holder was attached, the inlet shielded with a PTFE funnel, and indoor tubing insulated to prevent condensation from forming. The Aeris instruments solely used the DNPH-scrubbing method for zeroing."

Line 484 - Define / describe and/or cite the York regression (e.g., York et al...)

A reference to the York regression technique is provided.

Lines 157-159: "Data were averaged to 5 minutes and each regime fitted using a York regression (York et al., 2004) with standard deviations of the measurements used as uncertainty."

Line 486 to end of section - I'm assuming you are referring to York regression slopes and intercepts in this discussion, but that should be stated or clarified.

Ther description for Fig. 8 description has been expanded to note that the slopes and intercepts result from application of the York regression technique.

Lines 506-509: "Figure 8 – Comparison of ambient observations from the three HCHO monitors assessed in this work. (a)  Pico and G2307 observations taken at SDK in 2022 and 2023, (b) Pico and Ultra with 2022 measurements taken at GT in 2022 and SDK in 2023 (c) Ultra and G2307 observations at SDK 2023. Slopes and intercepts result from applying the York regression technique which incorporates the respective uncertainties of each instrument."

Figure 9 - The error bars on the TO-11A DNPH measurements seem way too small. You state earlier the uncertainty in the calibration standard is 15% (Line 367), so the total uncertainty in the measurement must be larger than this. As per the PAMS Technical Assistance Document, precision for collocated samples needs to be +/- 20%, so it's unlikely the uncertainty in the DNPH measurements will be below this. Double check the uncertainty in the DNPH measurements and correct the error bars accordingly.

We now ascribe an uncertainty of 20% to the DNPH measurements per section 13.4 of the TO-11A compendium. We have recalculated Fig. 9 accordingly and updates to the manuscript reflecting the new values can be found below:

Lines 369-371: "Method TO-11A requires that collocated DNPH-samples produce observations within 20 %, which is vindicated through EPA historical data (U.S. EPA, 1999). As such, an uncertainty of 20 % is assumed for TO-11A observations in this work."

Lines 514-516: "Fig. 9 compares G2307 observations from June-Aug. 2022 with those from co-located TO-11A measurements. 1 min integrated G2307 concentrations are averaged to the 8 h TO-11A sampling window. We find moderate correlation (r = 0.75) and a -52 % NMB of TO-11A observations relative to the G2307 (slope = 0.35 ± 0.02)."

Lines 534-535: "Figure 9 – 8 h TO-11A DNPH observations compared to Picarro G2307 observations at the SDK site from June through August 2022. Error bars represent the 10 % and 20 % uncertainty associated with the G2307 and TO-11A measurements."

[Figure]

**Figure 9 – 8 h TO-11A DNPH observations compared to Picarro G2307 observations at the SDK site from June through August 2022. Error bars represent the 10 % and 20 % uncertainty associated with the G2307 and TO-11A measurements.**

General Comment - A lot of abbreviations are used in this manuscript (HCHO, DR, MS, HO, ACN, NMB, etc.). I recommend the authors consider whether the paper is easier to read and understand using these abbreviations or if writing out some of the words would make the text easier to comprehend. This is a stylistic consideration the authors should think about, not a recommendation for one way or the other.

We appreciate the referee's suggestion. The acronym for acetonitrile has been removed as it only appears in one paragraph throughout the manuscript. Otherwise, we opt to keep the existing acronyms.

Lines 362-367: "The cartridges were then eluted with 10 mL of acetonitrile and the eluent analyzed via a Waters HPLC-UV system with a temperature stabilized (25 ± 1°C), reversed phase C18-coated silica gel (1.7 µm particle size) column (Bridged ethyl hybrid, 2.1 mm x 50 mm ID) at 360 nm wavelength. The eluents used in the HPLC process were deionized H2O and acetonitrile. The HPLC system was calibrated before each use with known concentrations of HCHO and field samples are analyzed in comparison to blank cartridges."

**Report #2:**

Just a few minor technical corrections after reading through the resubmitted manuscript:

Line 56: 'all of which a proton-transfer-reaction mass spectrometer should read 'all of which employ a proton-transfer-reaction mass spectrometer.'

> We thank the referee for catching this grammatical mistake and have updated the text accordingly.

> Lines 52-57: "While other studies have demonstrated the feasibility for continuous measurements via various spectroscopy-based methods (Yokelson et al., 1999; Cardenas et al., 2000; Dasgupta et al., 2005; Hak et al., 2005; Spinei et al., 2018; St Clair et al., 2019; Dugheri et al., 2021), the number of multi-month, ground-based, continuous, in-situ HCHO measurements is limited to a handful of studies, all of which employ a proton-transfer-reaction mass spectrometer (Warneke et al., 2013; Hansen et al., 2014; Coggon et al., 2021)."

Line 76: HDO line is at 2831.8413 cm-1 and not 2931.8413 cm-1.

> We've corrected the manuscript to include the appropriate value where the instrument searches for the HDO line.

> Lines 75-76: "The Aeris MIRA technique relies on a HDO line (located at 2831.8413 $cm^{-1}$) for spectral referencing."

Line 184: Typo in therefor.

> This grammatical error has been updated on lines 185-186: "Data was further screened to exclude points where scrubbers were exhausted and therefore unreliable."

Line 269: Missing word: Mention what remained consistently stable.

> This grammatical error has been corrected on lines 270-271: "We found 180 s to be the longest length of time between zeroes that either unit achieved where the instrument-reported HCHO signal remained consistently stable."

Line 404: It seems like there is better precision with the corrected Aeris Allan-Werle curves at longer integration times (Fig 4), so why does the text say 'lower precisions'?

> "Lower" in this context was used in a quantitative sense as it described the numerical values of precision. We have updated the manuscript to describe the precisions achieved by the instruments more qualitatively to avoid future confusion.

> Lines 407-408: "The corrected Aeris Allan-Werle curves trend similarly to the G2307's, achieving better precisions with longer integration times."

Fig 2: Mention what was being sampled in the caption of Figure 2. Was it ambient air?

We have updated the description in Fig. 2 to clarify what was being sampled.

Lines 209-210: "Figure 2 – Picarro G2307 baselines determined using the DR, DR+MS, or DNPH scrubbing methods. Each data point represents a consecutive, 4.5-min averaged DNPH and DR baseline measurement while sampling ambient air."

Fig 4: Some of the dashed lines are only partially dashed in subplots (b) and (c). Should not the whole line be dashed to correspond with the caption and text?

We're unsure in which manuscript version this error has been found. We have regardless double-checked Fig. 4 to make sure the plots are consistent in what they're describing.